# Mitigating the Modality Gap in Vision–Language Models with Fractal Spectral Geometry

Zihan Zhou[1]   Yang Zhou[1]   Ruoming Jin[2]   Pan He[1]   Patrick Emami[3]

## Abstract

Vision–language models such as CLIP embed images and text into a shared space, but still suffer from a modality gap, where image and text features cluster separately and nearest neighbors are dominated by same-modality rather than true cross-modal matches. Existing works alleviate the modality gap by strengthening cross-modal losses, post-processing embeddings or similarities, or imposing geometric regularization, but they primarily enforce global alignment and can distort local geometry, limiting gains in local ranking and zero-shot accuracy. We propose Fractal Spectral Alignment (FSAlign), which reduces the modality gap by shaping and matching the multi-scale geometry of image and text embeddings. By enforcing Ahlfors-regularity and sub-Gaussian heat kernel bounds, FSAlign constructs a shared fractal multi-scale structure for multiple modalities. This structure captures geometry across scales, from local neighborhoods to global structure, and ensures shared fractal spectral geometry across modalities. Based on this structure, we introduce a fractal spectral zeta score derived from multi-scale heat kernels and minimize the discrepancy between pairwise image–text samples to align their multi-scale neighborhoods. We theoretically demonstrate that FSAlign can guarantee the alignment of local spectral measures and global fractional Dirichlet energies. Our source code is available at https://github.com/zzz0134/FSAlign.

## 1. Introduction

Vision–language models (VLMs), such as CLIP (Radford et al., 2021), SigLIP (Zhai et al., 2023), and Open-CLIP (Cherti et al., 2023), are designed to learn a shared embedding space in which semantically matched image–text pairs are closely aligned while mismatched pairs are separated through contrastive learning on paired data. In recent years, CLIP and its variants (Zhang et al., 2024; Eslami & de Melo, 2024; Yamaguchi et al., 2025; Huang et al., 2025) have attracted substantial research interest for a range of multimodal tasks, including zero-shot image classification via prompting and text supervision transfer (Pan et al., 2025; Saha et al., 2024; Stojnić et al., 2024), as well as large-scale image–text retrieval using shared representations for cross-modal matching (Wang et al., 2024; Jing et al., 2024).

Despite achieving remarkable performance, CLIP and its variants continue to face a fundamental challenge known as the modality gap, where image and text embeddings occupy disjoint regions of the shared space rather than becoming fully overlapping. Consequently, samples within the same modality are often more semantically similar to each other than to their true cross-modal counterparts. Current research to mitigate this gap can be broadly categorized into three classes: (1) Training-time interventions, which employ stronger cross-modal constraints to enforce cross-modal consistency (Dong et al., 2025; Eslami & de Melo, 2025; Yaras et al., 2024; Sofer et al., 2025); (2) Post-training processing, which standardize embeddings or calibrate similarity scores at the representation or similarity level to reduce modality discrepancies (An et al., 2025; Yamashita et al., 2025); and (3) Geometric structure analysis, which analyzes the intrinsic structure of the embedding space to motivate alternative spaces or geometric regularizers (Ramasinghe et al., 2024; Levi & Gilboa, 2025; Schrodi et al., 2025).

However, two critical issues remain unresolved. (1) Global alignment does not guarantee local ranking: retrieval and matching are governed by local neighborhoods and ranked lists, and same modality samples may still dominate the nearest neighbors ahead of true cross-modal matches; and (2) Minimizing the modality gap does not necessarily translate to better downstream performance. In particular, it may increase the similarity among negative samples, compress-

[1]Auburn University, USA [2]Kent State University, USA [3]National Laboratory of the Rockies, USA. Correspondence to: Yang Zhou <yangzhou@auburn.edu>.

*Proceedings of the 43rd International Conference on Machine Learning*, Seoul, South Korea. PMLR 306, 2026. Copyright 2026 by the author(s).

ing ranking margins, potentially degrading retrieval and zero-shot accuracy even as the global gap diminishes.

To our best knowledge, this work is the first to leverage spectral theory and fractal geometry theory to mitigate the modality gap in VLMs by designing a shared fractal multi-scale structure for multiple modalities to capture the geometry from local to global scales with better performance. Our framework builds a shared fractal multi-scale structure for multiple modalities, aligning pairwise samples via spectral zeta functions derived from heat kernels across multiple diffusion scales. This design improves cross-modal retrieval and zero-shot classification by making pairwise samples share consistent spectral structure across diffusion scales, which preserves cross-modal nearest-neighbor rankings and keeps the global embedding structure well behaved. We formalize this by leveraging Ahlfors-regularity and sub-Gaussian heat kernel bounds to derive spectral geometric objectives, utilizing a zeta-based objective to bridge cross-modal alignment with spectral measures and their associated fractional Dirichlet energies.

Spacecraft rendezvous and docking is a classical multi-stage guidance problem (International Deep Space Standards, 2019). At long range, the objective is to steer the relative trajectory into a safe and controllable approach region, whereas at close range the goal shifts to minimizing relative pose and position errors near the docking port until physical docking becomes feasible. This multi-scale perspective motivates our connection between rendezvous control and the modality gap in VLMs. We do not only pull paired image and text embeddings closer. We shape the two modalities into a shared fractal multi scale structure. By aligning the spectral architecture of pairwise samples across these scales, we enforce consistent nearest-neighbor relationships across modalities from a viewpoint of local geometry. This multi-scale alignment effectively reduces the modality gap and significantly enhances performance on downstream tasks.

First, to align image and text neighborhoods across multiple scales, we construct a shared fractal multi-scale structure for multiple modalities. On fractal spaces satisfying Ahlfors-regularity and sub-Gaussian heat kernel bounds, the growth of neighborhood mass and the scaling of diffusion follow stable power-law across scales (Grigor'yan et al., 2009; Grigor'yan & Telcs, 2012). Given image and text embeddings, we enforce power-law growth of metric-ball mass over a log-spaced set of radii, which sets the fractal geometric dimension $d_f$. Simultaneously, we build a heat diffusion operator and heat kernel over log-spaced diffusion scales and constrain the heat trace to follow a power law in the diffusion scale, which defines the spectral geometric dimension $d_s$. Optimizing these constraints makes the two modalities share the same scaling pair $(d_f, d_s)$, yielding comparable geometric and diffusion behavior across scales.

Analogous to spacecraft rendezvous and docking, where relative motion must remain well behaved across distance scales prior to precision alignment, this shared fractal structure provides a common structural reference that facilitates robust cross-modal alignment.

Second, once the modalities share a common fractal structure, we align paired samples by matching their multi-scale diffusion neighborhoods across multiple diffusion scales. For each modality, we perform multi-scale diffusion over the embedding space. At each diffusion scale, we compute the heat value at each sample, which quantifies how strongly diffusion stays around that sample at the given scale. These heat values are then aggregated across scales into a spectral zeta score, which captures how the sample's neighborhood evolves as the diffusion scale increases. Alignment is achieved by matching this zeta score between paired image and text samples. The intuition parallels spacecraft rendezvous and docking, where relative behavior must remain well controlled across distance scales, from local neighborhoods to global structure, before precision matching. Under Ahlfors-regularity and sub-Gaussian heat kernel bounds, we theoretically derive that a small zeta-matching loss implies that pairwise samples possess consistent spectral measures across diffusion scales. Consequently, the same semantic concept induces similar neighborhood structures in both modalities, leading to more consistent nearest-neighbor rankings across image and text embeddings. This consistency mitigates same-modality bias, reduces the modality gap, and ultimately improves downstream task performance.

Figure 1 shows how our approach mitigates the modality gap and improves downstream retrieval performance. Figure 1 (a) gives an image–text pair, "a boy in a striped blue shirt and gray pants, holding a blue scooter, walking by a river." In the image embedding space, its local neighbors (blue) are mostly riverbanks and park scenery, while in the text embedding space, its neighbors (red) are mostly images of boys in blue shirts. The two neighborhoods exhibit minimal overlap. In CLIP, the correct image is ranked $166^{th}$ for text-to-image retrieval, and the correct caption is ranked $26^{th}$ for image-to-text retrieval. Figure 1(b) shows our training pipeline and the corresponding result for the same example. We first obtain separate latent embedding spaces for images and texts using their respective encoders. We then build a shared multi-scale fractal structure, enforcing matched fractal spectral geometric dimensions via ball-volume growth and heat-trace power-law constraints. Building on this shared structure, we align the fractal spectral descriptors of each image–text pair across multiple diffusion scales, ensuring that the multi-scale local geometry around corresponding samples is consistent across modalities. After training with these objectives, the correct image–text pair appears within the top three results in both retrieval directions, demonstrating that the local multi-scale

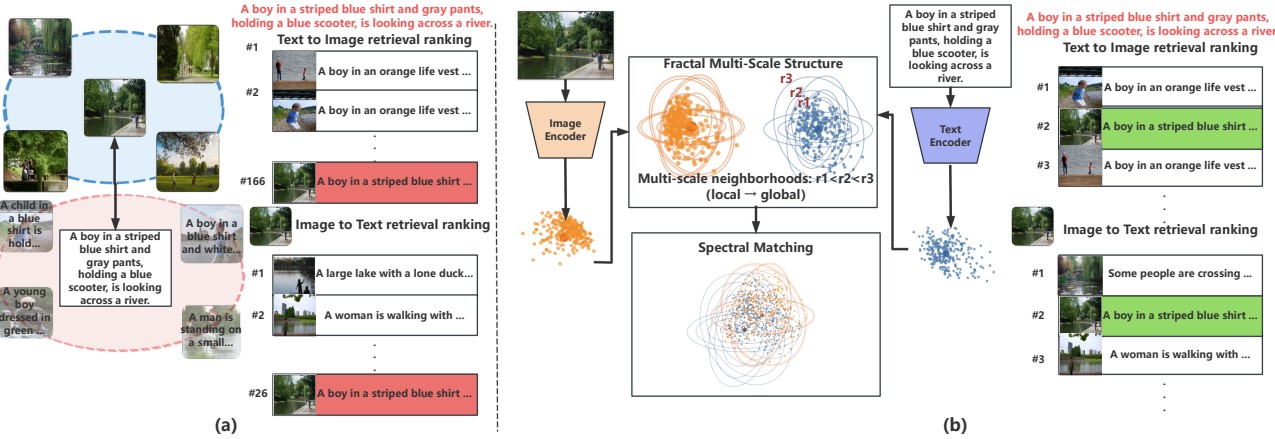

*Figure 1.* Local neighborhood inconsistency between image and text embeddings in CLIP (a) and its reduction by FSAlign (b).

geometry of image and text embeddings has been effectively aligned and that the modality gap is substantially reduced while preserving global alignment.

In comparison with existing approaches for mitigating the modality gap, our method offers three key advantages. (1) It reduces the gap by shaping image and text embedding spaces to follow a shared fractal multi-scale structure, ensuring that both modalities obey consistent multi-scale scaling laws rather than relying solely on pulling pairwise embeddings closer at only global scale in a latent space. (2) It improves cross-modal retrieval performance and strengthens zero-shot classification by matching the fractal spectral geometry of pairwise samples across multiple diffusion scales. This alignment promotes consistency in local neighborhood structures and nearest-neighbor rankings as well as global geometry between image and text embeddings, thereby reducing same-modality bias. (3) It establishes a theoretical connection between the proposed zeta-matching objective and cross-modal local structure. Under Ahlfors-regular volume growth and sub-Gaussian heat kernel conditions, we prove that our FSAlign method can guarantee the alignment of spectral measures and global fractional energies.

Empirical evaluation on real datasets demonstrates the superior performance of our fractal spectral alignment approach against several state-of-the-art modality gap mitigation methods on both text-to-image retrieval and zero shot classification. In addition, more experiments, implementation details, and hyperparameter selection and setting are presented in Appendices D.

## 2. Preliminary

In CLIP-style vision-language models, a set of paired samples $\{(I_i, T_i)\}_{i=1}^N$ is mapped into a shared embedding space as two sets of vectors. We denote the image embeddings by $\{x_i\}_{i=1}^N$ and the text embeddings by $\{y_i\}_{i=1}^N$, typically

$\ell_2$-normalized. The modality gap refers to the phenomenon that image and text embeddings still remain clearly separated in the embedding space. Even when two samples describe the same semantics, the paired image and text can have different nearest-neighbor relations within their own modalities. As a result, retrieval and zero-shot classification may favor same-modality neighbors over cross-modal matches.

To quantify this effect, a widely used metric is the centroid distance (CD) (Liang et al., 2022b).

$$CD = \|\frac{1}{N}\sum_{i=1}^N x_i - \frac{1}{N}\sum_{i=1}^N y_i\|_2, \quad (1)$$

which measures how far the two modality centers are from each other. A raw global distance can be influenced by how spread out the embeddings are within each modality. The relative modality gap (RMG) (Schrodi et al., 2025) therefore normalizes the average cross-modal distance by a within-modality reference scale. Let

$$D_{\text{pair}} = \frac{1}{N}\sum_{i=1}^N d(x_i, y_i),$$

$$D_{\text{intra}} = \frac{1}{2}\left(\frac{1}{N(N-1)}\sum_{i\neq j} d(x_i, x_j)\right.$$

$$\left. + \frac{1}{N(N-1)}\sum_{i\neq j} d(y_i, y_j)\right), \quad (2)$$

where $d(\cdot, \cdot)$ is a distance in the embedding space. Relative modality gap then uses the ratio

$$RMG = \frac{D_{\text{pair}}}{D_{\text{pair}} + D_{\text{intra}}}, \quad (3)$$

which reports cross-modal separation relative to the typical within-modality scale and is more comparable across datasets and training settings.

These global metrics capture overall separation but do not directly reflect ranking bias in retrieval. To measure local consistency, the normalized alignment score (NAS) (Naber et al., 2025) compares the overlap of $k$-nearest-neighbor sets. For each index $i$, let $\mathcal{N}_i^{\text{img}}(k)$ be the $k$ nearest neighbors of $x_i$ within $\{x_j\}_{j=1}^N$, and let $\mathcal{N}_i^{\text{txt}}(k)$ be the $k$ nearest neighbors of $y_i$ within $\{y_j\}_{j=1}^N$. Then

$$NAS(k) = \frac{1}{N} \sum_{i=1}^N \frac{\left| \mathcal{N}_i^{\text{img}}(k) \cap \mathcal{N}_i^{\text{txt}}(k) \right|}{k}, \qquad (4)$$

which reflects whether paired image and text samples have similar local neighborhoods in the two modalities.

Finally, the Cross-Modal Alignment Score (CMAS) (Eslami & de Melo, 2025) measures the alignment strength of paired samples by the average cosine similarity

$$CMAS = \frac{1}{N} \sum_{i=1}^N x_i^\top y_i, \qquad (5)$$

which directly reflects cross-modal similarity in the shared embedding space, and larger values indicate stronger pairwise alignment.

## 3. Fractal Multi-Scale Structure

Cross-modal retrieval and zero-shot classification are inherently dependent on local neighborhood topology and ranking integrity. Even when image and text embeddings are globally aligned, same-modality neighbors frequently persist in the top-ranked positions, causing the ground-truth cross-modal match to be ranked lower. In addition, excessively rigid alignment can inadvertently distort the relative geometry between positives and negatives, compressing the ranking margin and rendering the latent space less separable. As a result, the modality gap remains a persistent challenge in CLIP and its derivatives. To address these limitations, we introduce a structured multi-scale fractal geometric constraint based on Ahlfors-regularity and sub-Gaussian heat kernel bounds. For each modality, we impose the probability mass of metric balls to obey a power-law growth relative to the radius across a log-spaced range of scales, thereby enforcing consistent volume growth behavior. Simultaneously, we construct a heat-kernel-induced fractional diffusion operator over a log-spaced set of diffusion scales, constraining the heat trace to scale as a power law, which enforces consistent diffusion scaling behavior across modalities. This design ensures that the image and

text embedding spaces are comparable in their geometry and diffusion across scales. Analogous to spacecraft rendezvous and docking, where relative motion must remain well behaved across distance scales before precise alignment is feasible, the resulting shared fractal multi-scale structure provides a common structural reference for robust cross-modal alignment.

To construct this multi-scale fractal geometry, we first introduce Ahlfors regularity, sub-Gaussian heat kernels, and spectral dimension.

**Definition 1** (Ahlfors $d_f$-regularity). (Laakso, 2000) A metric-measure triple $(X, d, \mu)$ is Ahlfors $d_f$-regular if there exist constants $c_1, c_2 > 0$ and $r_0 > 0$ such that, for all $x \in X$ and $r \in (0, r_0]$,

$$c_1 r^{d_f} \le \mu\big(B_d(x, r)\big) \le c_2 r^{d_f}, \qquad (6)$$

where $X$ is a set with its Borel $\sigma$-algebra, $d : X \times X \to [0, \infty)$ is a metric, $\mu$ is a Borel probability measure on $X$, $B_d(x, r) = \{y \in X : d(x, y) \le r\}$ is the $d$-ball centered at $x$, and $d_f > 0$ is the geometric dimension.

**Definition 2** (Heat kernel). (Cao & Qiu, 2023) Let $L$ be a nonnegative self-adjoint operator on $L^2(X, \mu)$ and let $P_s = e^{-sL}$ have a jointly continuous kernel $p_s(x, y)$, so that $P_s f(x) = \int_X p_s(x, y) f(y) d\mu(y)$. We say $p_s$ is sub-Gaussian with walk dimension $d_w > 2$ if there exist constants $a_1, a_2, a_3, a_4 > 0$ such that, for all $s \in (0, 1]$ and $x, y \in X$,

$$\frac{a_1}{\mu\big(B_d(x, s^{1/d_w})\big)} \exp\left[ -a_2 \left( \frac{d(x,y)^{d_w}}{s} \right)^{\frac{1}{d_w-1}} \right] \le p_s(x, y)$$
$$\le \frac{a_3}{\mu\big(B_d(x, s^{1/d_w})\big)} \exp\left[ -a_4 \left( \frac{d(x,y)^{d_w}}{s} \right)^{\frac{1}{d_w-1}} \right], \qquad (7)$$

**Definition 3** (Spectral dimension). (Bañuelos & Baudoin, 2013) Let $\Theta_L(s) = \text{Tr}\big(e^{-sL}\big)$. The spectral dimension is $d_s > 0$ if, as $s \downarrow 0$, $\Theta_L(s) \asymp s^{-d_s/2}$. where $\asymp$ denotes two-sided bounds up to positive constants independent of $s$. When Definitions 1 and 2 hold one has $d_s = 2d_f/d_w$. Equivalently, for any $\alpha \in (0, 1]$,

$$\Theta_{L^\alpha}(s) = \text{Tr}\big(e^{-sL^\alpha}\big) \asymp s^{-d_s/(2\alpha)} \qquad . \qquad (8)$$

The following analysis specifies a multi-scale structural target for the image and text embedding spaces. Definitions 1–3 formalize power-law neighborhood growth, sub-Gaussian diffusion across scales, and the corresponding spectral scaling through the heat trace. Theorem 1 gives sufficient conditions under which a learned latent quadruple is a fractal metric-measure space with geometric dimension and spectral dimension. Appendix C proves that the learned

image and text latent spaces satisfy these conditions on the selected radii and diffusion scales, so both modalities share comparable neighborhood growth and diffusion behavior before cross-modal matching is applied.

**Theorem 1** (Fractal metric-measure latent space)*. Let $\mathcal{S} \subset (0, s_0]$ be a finite log-spaced set of diffusion scales and fix an order $\alpha \in (0,1]$. A quadruple $\mathcal{M}_\theta = (X, d_\theta, \mu, L_\theta)$, with $d_\theta : X \times X \to [0,\infty)$ a metric on $X$ and $L_\theta$ a nonnegative self-adjoint operator on $L^2(X, \mu)$, is a fractal metric-measure latent space if there exist $d_f > 0$, $c_1, c_2 > 0$, $r_0 > 0$, a walk dimension $d_w$, offsets $\{b_s\}_{s \in \mathcal{S}} \subset \mathbb{R}$, and a tolerance $\tau \in [0, \tau_0]$ such that:*

1. *$(X, d_\theta, \mu)$ is Ahlfors $d_f$-regular in the sense of Definition 1;*

2. *the fractional generator $(L_\theta)^\alpha$ generates a conservative symmetric Markov semigroup $\{e^{-s(L_\theta)^\alpha}\}_{s>0}$ whose kernel satisfies the sub-Gaussian bounds of Definition 2 with respect to $d_\theta$ and walk dimension $d_w$. We write*

$$K_\theta(s) = e^{-s(L_\theta)^\alpha}, \quad \Theta_\theta(s) = \mathrm{Tr} K_\theta(s); \quad (9)$$

3. *the heat trace $\Theta_\theta(s)$ realises the fractional spectral law up to tolerance $\tau$,*

$$\left| \log \Theta_\theta(s) + \tfrac{d_s}{2\alpha} \log s - b_s \right| \leq \tau, \quad d_s = \frac{2d_f}{d_w}. \quad (10)$$

*Under these conditions, $(X, d_\theta, \mu)$ is Ahlfors $d_f$-regular at radii $r \in (0, r_0]$, admits fractional heat operators $K_\theta(s)$ with walk dimension $d_w > 2$, and exhibits $\Theta_\theta(s) \asymp s^{-d_s/(2\alpha)}$ on the scales $\mathcal{S}$.*

We align the fractal latent spaces of the img and txt modalities. For each modality $m \in \{\mathrm{img}, \mathrm{txt}\}$ the encoder outputs latent points $X_m = \{z_i^{(m)}\}_{i=1}^{N_m}$ and a learned metric $d_\theta^m : X_m \times X_m \to [0,\infty)$. Metric balls are

$$B_{d_\theta^m}(x,r) = \{y \in X_m : d_\theta^m(y,x) \leq r\} (x \in X_m, r > 0). \quad (11)$$

Let $\mu_m$ be a probability measure on $X_m$. We use the uniform empirical measure on $X_m$, so for any $B \subseteq X_m$, $\mu_m(B) = \frac{|B|}{N_m}$. Choose log-spaced calibration radii $\mathcal{R} \subset (0, r_0]$ and set

$$\rho_{\max} = \min_{r \in \mathcal{R}} \frac{r_0}{r}, \quad \Lambda \subset (1, \rho_{\max}], \quad (12)$$

so that for every $r \in \mathcal{R}$ and every $\rho \in \Lambda$ we have $\rho r \leq r_0$. For diffusion scales take $\mathcal{S} \subset (0, s_0]$ log-spaced and fix an order $\alpha \in (0,1]$. For each modality $m \in \{\mathrm{img}, \mathrm{txt}\}$, let $L_\theta^m$ be a nonnegative self-adjoint operator on $L^2(X_m, \mu_m)$ constructed from $d_\theta^m$. With spectral resolution $L_\theta^m = \int_0^\infty \lambda dE_\theta^m(\lambda)$, define

$$\left(L_\theta^m\right)^\alpha = \int_0^\infty \lambda^\alpha dE_\theta^m(\lambda), \quad (13)$$

and set

$$K_\theta^m(s) = \exp\left(-s\left(L_\theta^m\right)^\alpha\right), \quad \Theta_\theta^m(s) = \mathrm{Tr} K_\theta^m(s). \quad (14)$$

To shape the geometric part of the fractal, we make ball mass grow as a power of the radius, consistent with the Ahlfors law at exponent $d_f$: a ball of radius $r$ should contain about $r^{d_f}$ mass. We enforce this by penalising a symmetric relative discrepancy between $\mu_m\left(B_{d_\theta^m}(x, \rho r)\right)$ and $\lambda^{d_f} \mu_m\left(B_{d_\theta^m}(x,r)\right)$:

$$\mathcal{L}_{\mathrm{dbl}}(\theta, d_f) = \frac{1}{2} \sum_{m \in \{\mathrm{img},\mathrm{txt}\}} \mathbb{E}_{x \sim \mu_m} \frac{1}{|\mathcal{R}||\Lambda|} \sum_{r \in \mathcal{R}} \sum_{\rho \in \Lambda}$$
$$\left( \frac{\mu_m\left(B_{d_\theta^m}(x, \rho r)\right) - \rho^{d_f} \mu_m\left(B_{d_\theta^m}(x,r)\right)}{\mu_m\left(B_{d_\theta^m}(x, \rho r)\right) + \rho^{d_f} \mu_m\left(B_{d_\theta^m}(x,r)\right)} \right)^2. \quad (15)$$

On the spectral side, the heat trace follows a power law in the diffusion scale with exponent $d_s$ and order $\alpha$. Writing $\mathcal{S} = \{s_1 < \cdots < s_K\}$, we match the ratios of heat traces across adjacent diffusion scales to the corresponding power-law ratios:

$$\mathcal{L}_{\mathrm{spec}}(\theta, d_s) = \frac{1}{2(K-1)} \sum_{m \in \{\mathrm{img},\mathrm{txt}\}} \sum_{k=1}^{K-1}$$
$$\left( \frac{\mathrm{Tr}\left( \exp\left(-s_{k+1}(L_\theta^m)^\alpha\right)\right)}{\mathrm{Tr}\left( \exp\left(-s_k(L_\theta^m)^\alpha\right)\right)} - \left(\frac{s_{k+1}}{s_k}\right)^{-\frac{d_s}{2\alpha}} \right)^2. \quad (16)$$

We use the sum of the two losses as the training objective:

$$\mathcal{L}_{\mathrm{frac}}(\theta, d_f, d_s) = \mathcal{L}_{\mathrm{dbl}}(\theta, d_f) + \mathcal{L}_{\mathrm{spec}}(\theta, d_s) \quad (17)$$

Minimizing $\mathcal{L}_{\mathrm{frac}}$ makes the img and txt latent spaces share the same $(d_f, d_s)$ on the chosen radii $\mathcal{R}$ and diffusion scales $\mathcal{S}$.

In words, the two loss terms turn the learned latent spaces into fractal metric-measure spaces. For each modality $m \in \{\mathrm{img}, \mathrm{txt}\}$ we consider the latent quadruple $\mathcal{M}_\theta^m = (X_m, d_\theta^m, \mu_m, L_\theta^m)$; in Appendix X we show that these quadruples satisfy the regularity conditions of Theorem 1. Thus, for any target dimensions $(d_f, d_s)$ and any tolerance $\tau \in [0, \tau_0]$, if the training objective $\mathcal{L}_{\mathrm{frac}}(\theta, d_f, d_s)$ in (17) satisfies $\mathcal{L}_{\mathrm{frac}}(\theta, d_f, d_s) \leq \tau^2$, then each $\mathcal{M}_\theta^m$ is a fractal metric-measure latent space in the sense of Theorem 1, with geometric dimension $d_f$ and spectral dimension $d_s$.

# 4. Fractal Spectral Alignment

In multimodal contrastive learning, the modality gap refers to the phenomenon where image and text embeddings occupy disjoint regions of the shared space rather than becoming fully overlapping (Liang et al., 2022a). Semantic neighborhood analyses further reveal that, for a given concept, the nearest-neighbor sets in the image and text modalities exhibit minimal overlap. In many cases, the modality from which a representation is derived—image or text—better predicts its nearest neighbors than its semantic label (Naber et al., 2025). Consider the pairwise samples illustrated in Figure 1. A standard VLM yields embeddings $z_i^{\text{img}}$ and $z_i^{\text{txt}}$. In a small $k$-nearest-neighbor set around $z_i^{\text{img}}$, the image neighborhood is dominated by riverbank and park scenery images, whereas the text neighborhood primarily consists of images depicting boys wearing blue shirts. The overlap between these two sets is negligible, despite their representing the same semantic sample. This observation highlights that effective cross-modal alignment requires more than minimizing the distance between $z_i^{\text{img}}$ and $z_i^{\text{txt}}$; it also requires aligning the neighborhoods they induce across diffusion scales. The central question is whether these neighborhoods exhibit comparable multi-scale structures and can be synchronized within the shared space. Building on the fractal latent spaces established in Section 3, this section employs fractional heat diffusion to construct a local multi-scale spectral descriptor, enforcing its cross-modal consistency as a primary alignment objective.

For each modality $m \in \{\text{img}, \text{txt}\}$, Section 3 constructs a fractal metric-measure latent space $\mathcal{M}_\theta^m = (X_m, d_\theta^m, \mu_m, L_\theta^m)$, where $X_m = \{z_i^{(m)}\}_{i=1}^N$ is the latent point set for paired training samples, $d_\theta^m$ is the learned metric, and $\mu_m$ is the uniform empirical measure on $X_m$, $\mu_m(B) = |B|/N$ for any $B \subseteq X_m$. Let $\mathcal{S} = \{s_1 < \cdots < s_K\} \subset (0, s_0]$ be a log-spaced collection of diffusion scales and fix an order $\alpha \in (0, 1]$. For each $s \in \mathcal{S}$, define the fractional heat operator

$$K_\theta^m(s) = \exp\big(-s(L_\theta^m)^\alpha\big). \qquad (18)$$

To relate diffusion to neighborhoods in a way that matches this semigroup assumption, we represent $K_\theta^m(s)$ by an integral kernel on the measure space $(X_m, \mu_m)$. Since $K_\theta^m(s) : L^2(X_m, \mu_m) \to L^2(X_m, \mu_m)$ is bounded and $X_m$ is finite, there exists a measurable function $p_\theta^m(s; \cdot, \cdot)$ on $X_m \times X_m$ such that for any $f \in L^2(X_m, \mu_m)$ and any $x \in X_m$,

$$(K_\theta^m(s)f)(x) = \int_{X_m} p_\theta^m(s; x, y) f(y) d\mu_m(y). \qquad (19)$$

Mass preservation $K_\theta^m(s)\mathbf{1} = \mathbf{1}$ is equivalent to

$$\int_{X_m} p_\theta^m(s; x, y) d\mu_m(y) = 1, \quad \forall x \in X_m, \forall s > 0. \qquad (20)$$

For the paired samples $X_m = \{z_i^{(m)}\}_{i=1}^N$, we introduce the associated probability mass function

$$\pi_\theta^m(s; i, j) = p_\theta^m(s; z_i^{(m)}, z_j^{(m)}) \mu_m(\{z_j^{(m)}\}), \qquad (21)$$

so that $\sum_{j=1}^N \pi_\theta^m(s; i, j) = 1$ for all $i$ and $s > 0$. In particular, $\pi_\theta^m(s; i, \cdot)$ is a diffusion neighborhood distribution around $z_i^{(m)}$ at scale $s$.

We aggregate these neighborhood distributions across diffusion scales using a truncated Mellin-type descriptor on $(0, s_0]$, which matches the scale range used in training. Fix $q > \frac{d_s}{2\alpha}$. For modality $m$ and anchor index $i$, define the truncated local zeta profile

$$\zeta_i^m(q; j) = \frac{1}{\Gamma(q)} \int_0^{s_0} s^{q-1} \pi_\theta^m(s; i, j) ds, \quad j \in \{1, \ldots, N\}. \qquad (22)$$

Using the discrete scales $\mathcal{S} = \{s_k\}_{k=1}^K$, we approximate (22) by attaching positive quadrature weights $w_1, \ldots, w_K$ to $\{s_k\}_{k=1}^K$ and defining

$$\widehat{\zeta}_\theta^m(i; j) = \frac{1}{\Gamma(q)} \sum_{k=1}^K w_k s_k^{q-1} \pi_\theta^m(s_k; i, j), \quad j \in \{1, \ldots, N\}. \qquad (23)$$

The training data consist of paired samples $(z_i^{\text{img}}, z_i^{\text{txt}})$ with index set $\mathcal{I} = \{1, \ldots, N\}$. We match the local multi-scale descriptors by minimizing

$$\mathcal{L}_{\text{match}}(\theta) = \frac{1}{N} \sum_{i=1}^N \left\| \widehat{\zeta}_\theta^{\text{img}}(i; \cdot) - \widehat{\zeta}_\theta^{\text{txt}}(i; \cdot) \right\|_2^2, \qquad (24)$$

where $\widehat{\zeta}_\theta^m(i; \cdot)$ is the length-$N$ vector with entries $\{\widehat{\zeta}_\theta^m(i; j)\}_{j=1}^N$.

The fractal structural loss from Section 3 is

$$\mathcal{L}_{\text{frac}}(\theta, d_f, d_s) = \mathcal{L}_{\text{dbl}}(\theta, d_f) + \mathcal{L}_{\text{spec}}(\theta, d_s), \qquad (25)$$

with $\mathcal{L}_{\text{dbl}}$ and $\mathcal{L}_{\text{spec}}$ given in (15)-(17). It enforces a common geometric and spectral scaling pair $(d_f, d_s)$ across modalities. After this shared backbone is in place, reducing $\mathcal{L}_{\text{match}}$ aligns the induced multi-scale diffusion neighborhoods of paired samples.

We theoretically show that a small matching loss $L_{\text{match}}$ increases the cross-modal consistency of the Top-$k$ neighbor sets, quantified by $\text{NAS}(k)$, for most paired samples.

**Theorem 2** (Multi-scale diffusion neighborhoods imply higher cross-modal NAS). *Assume that $\mathcal{M}_\theta^{\text{img}}$ and $\mathcal{M}_\theta^{\text{txt}}$ satisfy Ahlfors $d_f$-regularity and sub-Gaussian heat kernel bounds with walk dimension $d_w > 2$, and that $(L_\theta^m)^\alpha$ generates a conservative symmetric Markov semigroup for each $m \in \{\text{img}, \text{txt}\}$. Fix $q > \frac{d_s}{2\alpha}$ and define the multi-scale*

*diffusion neighborhood descriptor $\widehat{\zeta}_\theta^m(i; \cdot)$ by (23), and the matching loss $\mathcal{L}_{\text{match}}$ by (24).*

*For any $\delta \in (0, 1]$, if $\mathcal{L}_{\text{match}}(\theta) \leq \varepsilon^2$, then for at least $(1 - \delta)N$ indices $i \in \mathcal{I}$ the following holds for every $k \geq 1$. Let $\mathcal{N}_k^m(i)$ be the indices of the $k$ largest entries of $\widehat{\zeta}_\theta^m(i; \cdot)$ (ties are broken deterministically), and let $\gamma_{i,k} > 0$ be the minimum, over $m \in \{\text{img}, \text{txt}\}$, of the gap between the $k$-th and $(k + 1)$-th largest entries of $\widehat{\zeta}_\theta^m(i; \cdot)$. Then the neighborhood agreement score $\text{NAS}_k(i)$ defined in Definition 4 satisfies*

$$\text{NAS}_k(i) \geq 1 - \frac{4}{k} \cdot \frac{1}{\gamma_{i,k}^2} \cdot \frac{\varepsilon^2}{\delta}. \quad (26)$$

*In particular, if $\varepsilon^2/\delta < \gamma_{i,k}^2/4$, then $\mathcal{N}_k^{\text{img}}(i) = \mathcal{N}_k^{\text{txt}}(i)$ and $\text{NAS}_k(i) = 1$.*

Finally, the training objective combines the structural and matching terms,

$$\mathcal{L}_{\text{total}}(\theta; d_f, d_s) = \mathcal{L}_{\text{frac}}(\theta, d_f, d_s) + \mathcal{L}_{\text{match}}(\theta). \quad (27)$$

The first term enforces a common fractal geometry within each modality; the second aligns the local multi-scale diffusion neighborhoods across modalities on that shared geometric backbone. Please refer to Appendix C for detailed proof of Theorem 2

## 5. Experimental Evaluation

**Datasets and Models.** In this section, we evaluate the effectiveness of our method compared to several state-of-the-art modality gap mitigation baselines. Through comprehensive experiments on multiple representative cross-modal retrieval and zero-shot classification tasks, we demonstrate that our method consistently improves performance across diverse settings. Please refer to the appendix F for detailed experimental settings and additional results.

We conduct cross-modal retrieval experiments on two widely used image-text benchmarks, Flickr30k (Young et al., 2014) and MS-COCO (Lin et al., 2015), using the standard Karpathy style splits for fair comparison. For zero-shot classification, we evaluate on commonly used image classification datasets including CIFAR-100 (Krizhevsky, 2009), Tiny ImageNet (TI-200) (Le & Yang, 2015), and DTD (Cimpoi et al., 2014) where class names are converted into textual prompts following the standard CLIP practice. We extract image/text representations using pretrained dual encoder backbones, including CLIP (Radford et al., 2021), SigLIP (Zhai et al., 2023), and OpenCLIP (Cherti et al., 2023), and apply our method as a learnable modality specific geometry layer on top of the frozen backbone embeddings. The detailed descriptions of the datasets and models are presented in Appendix.

**Baselines.** We compare our method with nine baselines under the same retrieval and zero-shot classification settings. CLIP (Radford et al., 2021),SigLIP (Zhai et al., 2023), and OpenCLIP (Cherti et al., 2023) are used as pretrained backbone baselines. IOT (An et al., 2024) reduces the modality gap by standardizing image and text embeddings within each modality before computing similarity. INCL (Mistretta et al., 2025) maps an input to an inverse-modality representation via modality inversion and evaluates similarity in the resulting cross-modal space. OTCL (Role et al., 2025) reduces the modality gap through procedures based on optimal transport. MG (Liang et al., 2022b) analyzes how contrastive learning induces a persistent gap and introduces training controls that adjust the gap behavior. ALIGNCLIP (Eslami & de Melo, 2024) improves cross-modal alignment by sharing learnable parameters between the modality encoders and optimizing a semantically regularized separation objective on the uni-modal embeddings. GR-CLIP (Li et al., 2025) reduces the gap by mean centering the text and image embeddings.

**Evaluation Metrics.** We evaluate performance on cross-modal retrieval, zero-shot classification, and modality-gap reduction. For retrieval on MS-COCO and Flickr30k, we report Recall@K with K $\in \{1, 5, 10\}$ for both image-to-text and text-to-image retrieval. For zero-shot classification on CIFAR-100, Tiny ImageNet-200, and DTD, we report top-1 accuracy using the standard prompt based classifier constructed from the text encoder. To quantify modality gap reduction, we report the four metrics defined in the preliminary, including centroid distance (CD), relative modality gap (RMG), neighborhood alignment score (NAS@10), and cross-modal alignment score (CMAS), and use the same distance and similarity definitions across all methods and backbones.

**Ablation study.** Fig. 4 compares FSAlign with two ablations. FSAlign-F keeps the multi scale fractal constraint but drops local zeta matching, and learns the geometry maps using the standard image–text contrastive loss on paired samples. FSAlign-M keeps local zeta matching but drops the multi scale fractal constraint, and it learns the geometry mapping as a plain projection on top of the backbone embeddings using only the zeta loss. FSAlign is the only variant that improves retrieval R@1 on both MS COCO and Flickr30k while also giving the best top 1 accuracy on CIFAR 100, TI 200, and DTD. Removing either term lowers retrieval and also lowers zero shot accuracy, with the clearest drop on TI 200, which matches the role of the fractal constraint in stabilizing neighborhood structure across radii and the role of zeta matching in tying each image to its paired text.

**Evaluation of retrieval performance.** Table 1 reports image-to-text and text-to-image Recall@1/5/10 on the

*Table 1.* Cross-modal retrieval results on the MS-COCO and Flickr30k Karpathy test splits with CLIP

| Method | MS-COCO | | | | | | Flickr30k | | | | | |
| | Image→Text | | | Text→Image | | | Image→Text | | | Text→Image | | |
| | R@1 | R@5 | R@10 | R@1 | R@5 | R@10 | R@1 | R@5 | R@10 | R@1 | R@5 | R@10 |
|---|---|---|---|---|---|---|---|---|---|---|---|---|
| CLIP | 40.4 | 64.66 | 73.96 | 27.32 | 50.25 | 61.08 | 69.40 | 90.10 | 94.30 | 56.14 | 80.24 | 86.86 |
| I0T | 44.22 | 69.16 | 78.66 | 29.04 | 52.72 | 63.95 | 72.60 | 91.50 | 95.90 | 56.42 | 80.62 | 87.78 |
| INCL | 40.78 | 65.98 | 76.06 | 28.80 | 54.03 | 64.63 | 66.10 | 89.20 | 94.70 | 55.90 | 81.70 | 88.26 |
| OTCL | 49.18 | 73.32 | 82.30 | 30.25 | 54.94 | 65.71 | 76.90 | 94.30 | 97.80 | 57.90 | 82.98 | 89.06 |
| MG | 34.04 | 58.54 | 69.76 | 23.27 | 45.54 | 55.86 | 62.90 | 88.10 | 94.50 | 48.40 | 73.44 | 81.30 |
| ALIGNCLIP | 39.76 | 66.62 | 77.30 | 25.83 | 51.62 | 63.50 | 68.8 | 89.70 | 93.70 | 49.74 | 77.24 | 85.50 |
| GR-CLIP | 43.44 | 68.72 | 78.78 | 30.49 | 54.52 | 65.58 | 73.30 | 93.30 | 96.40 | 58.10 | 82.34 | 88.86 |
| **FSAlign** | 51.26 | 75.30 | 83.82 | 31.20 | 56.00 | 66.90 | 78.30 | 95.23 | 98.20 | 59.80 | 83.64 | 90.02 |
| FSAlign-F | 45.56 | 69.26 | 78.70 | 30.36 | 54.28 | 64.79 | 75.00 | 93.30 | 96.60 | 57.80 | 82.08 | 88.96 |
| FSAlign-M | 43.18 | 67.08 | 76.76 | 29.39 | 52.71 | 63.41 | 73.80 | 91.80 | 95.90 | 58.56 | 82.48 | 88.72 |

*Table 2.* Zero-shot image classification results with CLIP

| Method | CIFAR-100 | | TI-200 | | DTD | |
| | Top-1 | Top-5 | Top-1 | Top-5 | Top-1 | Top-5 |
|---|---|---|---|---|---|---|
| CLIP | 62.62 | 87.03 | 40.31 | 72.55 | 55.79 | 80.23 |
| I0T | 63.28 | 88.28 | 41.01 | 73.08 | 56.93 | 80.40 |
| INCL | 62.48 | 87.31 | 40.70 | 71.36 | 55.28 | 80.10 |
| OTCL | 62.82 | 89.23 | 41.58 | 71.28 | 56.18 | 81.23 |
| MG | 53.12 | 74.47 | 34.68 | 66.54 | 46.93 | 70.92 |
| ALIGNCLIP | 52.09 | 77.98 | 27.97 | 56.48 | 41.08 | 66.41 |
| GR-CLIP | 63.20 | 87.85 | 41.38 | 73.13 | 56.48 | 80.32 |
| **FSAlign** | 66.28 | 90.0 | 51.85 | 75.78 | 59.23 | 83.12 |
| FSAlign-F | 62.77 | 87.74 | 47.81 | 80.98 | 55.95 | 80.69 |
| FSAlign-M | 63.08 | 86.28 | 49.20 | 81.47 | 56.40 | 80.36 |

*Table 3.* Modality gap metrics on MS-COCO with CLIP

| Method | CD | RMG | NAS@100 | CMAS |
|---|---|---|---|---|
| CLIP | 0.8876 | 0.5584 | 0.3428 | 0.2974 |
| I0T | 0.0316 | 0.4546 | 0.4246 | 0.3053 |
| INCL | 0.7623 | 0.5318 | 0.3043 | 0.2719 |
| OTCL | 0.8887 | 0.5626 | 0.2589 | 0.2732 |
| MG | 0.2504 | 0.4650 | 0.1416 | 0.4119 |
| ALIGNCLIP | 0.3474 | 0.4768 | 0.3824 | 0.5924 |
| GR-CLIP | 0.024 | 0.4551 | 0.3803 | 0.3036 |
| **FSAlign** | 0.7729 | 0.4922 | 0.4584 | 0.6490 |
| FSAlign-F | 0.8344 | 0.5297 | 0.3771 | 0.3503 |
| FSAlign-M | 0.8287 | 0.5626 | 0.2589 | 0.2731 |

Karpathy test splits for CLIP, SigLIP, and OpenCLIP. On both datasets, our method improves retrieval across all three backbones. The improvement is especially clear at Recall@1, where more queries place the correct paired item at the top of the ranked list instead of being preceded by same modality neighbors. For example with CLIP on MS-COCO, image-to-text Recall@1 increases from 40.4 to 51.3 and text-to-image Recall@1 increases from 27.3 to 31.2, with consistent improvements at Recall@5 and Recall@10. On Flickr30k, image-to-text Recall@1 increases from 69.4 to 78.3 and text-to-image Recall@1 increases from 56.1 to 59.8. Compared with existing modality gap mitigation baselines, our method achieves the strongest retrieval performance in most settings.

**Evaluation of modality gap metrics.** Table 3 reports four gap measures on MS-COCO. FSAlign achieves the best NAS@100 and CMAS, meaning the paired text appears more often among an image's nearest neighbors and the paired cosine similarity is higher. CD is not the smallest because we do not force the two modality means to coincide,

and retrieval is driven by local neighbor order rather than center distance alone. Both FSAlign-F and FSAlign-M drop on NAS@100 and CMAS, consistent with the retrieval and zero-shot ablations.

**Fractal multi scale scaling.** Fig. X(a–b) checks whether neighborhood growth follows a power law using ball mass $\mu(r)$ and the correlation integral $C(r)$ (Grassberger & Procaccia, 1983; Falconer, 1990). We fit straight lines in $(\log r, \log \mu(r))$ and $(\log r, \log C(r))$ on an automatically selected radius window $[r_{\min}, r_{\max}]$, and report the slopes as $d_f$ and $d_2$ with fit quality $R^2$. We also report the window width $W_r = \log(r_{\max}/r_{\min})$ and the within-window slope fluctuation $\sigma_{\mathrm{eff}}$, and the insets show local slope histograms. We use the same fitted window for CLIP and ours, with $W_r = 0.21$ in (a) and 0.22 in (b). On this fixed range, ours stays closer to a single line, most noticeably on text embeddings. Ball mass gives $R^2 = 0.996/0.993$ (image/text) compared with $0.980/0.945$ for CLIP. The correlation integral gives $R^2 = 0.992/0.995$ compared with $0.984/0.957$, and $\sigma_{\mathrm{eff}}$ decreases from $4.62/5.77$ to $1.74/2.39$. Overall, compared to CLIP, our representations follow the power law more tightly on the same radii, the neighborhood statistics

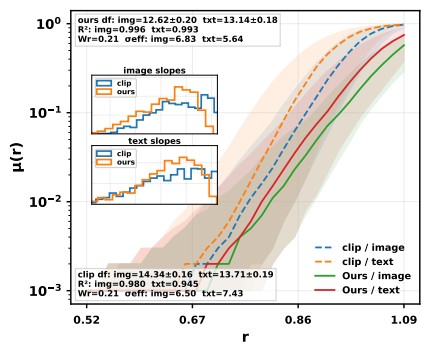

*Figure 2.* Ball-mass scaling.

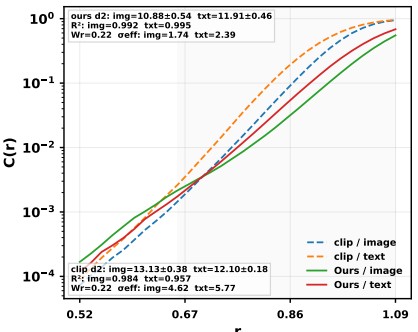

*Figure 3.* Correlation integral scaling.

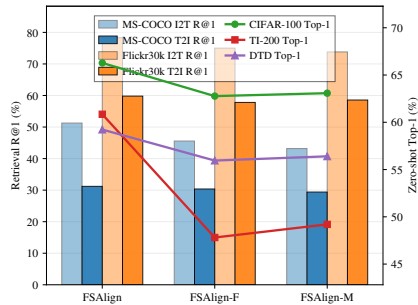

*Figure 4.* Ablation of FSAlign Variants on Retrieval and Zero-shot Classification.

CDF of local neighborhood overlap on flickr30k

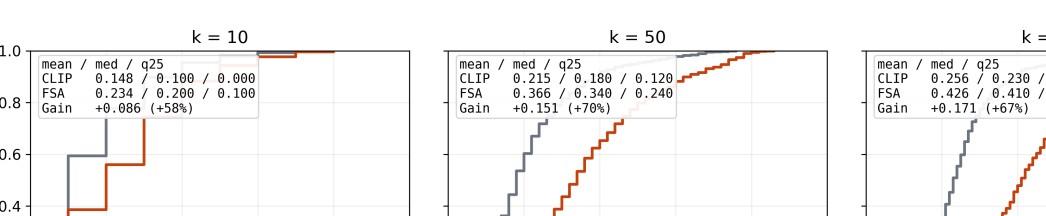

*Figure 5.* Dataset-level CDF of local neighborhood overlap on Flickr30k.

vary less across scales, and the scaling behavior is more consistent between image and text, so the embeddings indeed form a fractal multi-scale neighborhood growth pattern. Detailed settings are in Appendix F.

**Evaluation of zero-shot classification performance.** Table 2 reports top-1 and top-5 accuracy on CIFAR-100, TI-200, and DTD using the same prompt templates for all methods. FSAlign gives the best results on all three datasets. TI-200 shows the largest gain, where FSAlign improves top-1 from 40.31 to 51.85 and top-5 from 72.55 to 75.78. Multi-scale fractal constraints make same-class neighborhoods tighter and different-class neighborhoods more separated across radii, so under the same templates the correct class prompt is ranked higher and zero-shot accuracy increases.

**Effect of FSAlign on Local Neighborhood Structure** To directly examine whether FSAlign changes the local geometry of the shared space, Figure 5 visualizes the dataset-level distribution of local neighborhood overlap on Flickr30k. For each paired image-text sample, we compare the top-$k$ neighbors of the image in image space with the top-$k$ neighbors of its paired text in text space after mapping image neighbors to their paired texts. FSAlign shifts the full-sample distribution toward larger overlap values at $k = 10$, 50, and 100. For $k = 10$, the mean/median/$q_{25}$ increase from $0.148/0.100/0.000$ to $0.234/0.200/0.100$.

For $k = 50$, they increase from $0.215/0.180/0.120$ to $0.366/0.340/0.240$. For $k = 100$, they increase from $0.256/0.230/0.190$ to $0.426/0.410/0.320$. This shows that the improvement is not limited to a few isolated examples, but reflects a systematic increase in local cross-modal neighborhood consistency across the test set.

## 6. Conclusion

In this work, we proposed a novel fractal spectral alignment framework for mitigating the modality gap in vision-language models. First, we leverage fractal multi-scale geometry to construct a shared structural space for image and text embeddings, enforcing consistent neighborhood-growth and diffusion-scaling behavior across modalities. Second, we employ a local spectral matching objective to align paired image-text samples through their multi-scale diffusion neighborhoods, improving local ranking consistency and reducing the effect of strong local competitors. Finally, we theoretically analyze the proposed objective and establish its connection to the alignment of local spectral measures and fractional Dirichlet energies, providing guarantees on its effectiveness in reducing the modality gap. Extensive experiments verify that FSAlign consistently improves retrieval and zero-shot classification performance.

## Acknowledgements

This research is partially sponsored by the National Science Foundation (NSF) under Grant No. OAC-2313191.

## Impact Statement

In this work, we use public vision language benchmarks and standard evaluation protocols to study modality gap mitigation. We evaluate cross modal retrieval on the Karpathy test splits of MS COCO and Flickr30k, and evaluate zero shot image classification on CIFAR 100, Tiny ImageNet 200, and DTD. These datasets are widely used in training and evaluating representation learning systems. We describe the full training setup, prompt templates, and evaluation scripts in the appendix so the experiments can be reproduced on top of the same pretrained backbones.

The goal of this work is to reduce the gap between image and text representations without changing the downstream evaluation interface. We introduce FSAlign, which keeps the pretrained vision-language encoders fixed and learns two modality-specific geometry mappings on top of the frozen image and text embeddings, trained with our fractal neighborhood-growth loss, spectral scaling loss, and the paired local matching loss. The fractal constraint controls how neighborhood mass grows over radii inside each modality, and the local matching term aligns the cross modal structure at paired samples through a discrete zeta based descriptor. This design is applied on CLIP, SigLIP, and OpenCLIP under the same retrieval and zero shot protocols used by prior gap mitigation baselines.

Our results show that the same geometric constraint improves both ranking based retrieval and zero shot classification. On retrieval, more queries place the paired item at the top of the ranked list on both MS COCO and Flickr30k. On zero shot classification, applying our learned geometry maps to the frozen image and text embeddings improves top-1 and top-5 accuracy on CIFAR-100, Tiny-ImageNet-200, and DTD under the same prompt templates. The ablations further show that removing either the multi scale constraint or the local matching term consistently weakens both retrieval and zero shot accuracy, which supports the role of each component.

We hope this work provides a practical way to analyze and reduce the modality gap in modern vision–language models with a lightweight geometry-mapping training recipe. The method keeps the backbone architecture unchanged and trains only lightweight geometry maps on top of frozen embeddings, which makes it easy to test across different pretrained models and to integrate into existing evaluation pipelines. We expect these findings to be useful for research on multimodal retrieval and transfer, and for future work that connects local geometry across scales with cross modal alignment objectives.

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

# A. Related Work

Recent research efforts have focused on understanding and reducing the modality gap between embeddings of different data types to enhance the performance of multimodal learning (Zhang et al., 2013; Lee et al., 2013; Su et al., 2013; Zhang et al., 2014; Su et al., 2015; Zhou et al., 2015b; Bao et al., 2015; Zhou et al., 2015c; Lee et al., 2015; Zhou et al., 2015a; Zhou & Liu, 2015; Jiang et al., 2016; Zhou et al., 2016; 2018b;a; Palanisamy et al., 2018; Lee et al., 2019; Qu et al., 2020; Dong et al., 2020; Goswami et al., 2020; Zhang et al., 2021a; Zhao et al., 2021; Zhou et al., 2021; Ren et al., 2021; Zhang et al., 2021b; Yin et al., 2021; Zhou et al., 2022a; Guimu Guo & Zhou, 2022; Jin et al., 2022a; Zhang et al., 2022a; Che et al., 2022; Yan et al., 2022a; Liu et al., 2022; Yan et al., 2022b;c; Zhou et al., 2022d; Jin et al., 2022b; Che et al., 2023b; Hong et al., 2023; Chen et al., 2023; Che et al., 2023a; Liu et al., 2023; Ren et al., 2023; Liu et al., 2024c;b; Yuan et al., 2024b;a; ?; Zhou et al., 2024; Li et al., 2024; Liu et al., 2024e;d;a; Cheng et al., 2025; Yuan et al., 2025b; Jia et al., 2025; Liu et al., 2025; Che et al., 2025; Yuan et al., 2025a;c; Zhou et al., 2025).

## A.1. CLIP Style Contrastive Vision Language Models

Contrastive vision language pretraining has become the main learning scheme for vision language models (VLMs). With this scheme, VLMs can be applied to downstream tasks such as zero-shot classification and cross-modal retrieval. CLIP maps images and texts into a shared embedding space using an image encoder and a text encoder. Its contrastive objective pulls matched image-text pairs together and pushes mismatched pairs apart. As a result, both zero-shot classification and image-text retrieval can be implemented as similarity search in this shared space (Radford et al., 2021). ALIGN showed that scaling contrastive pretraining with much larger and noisier web pairs can further improve zero-shot transfer and cross-modal retrieval (Jia et al., 2021). Later work improved CLIP's objective design and scalability (Yu et al., 2022; Cherti et al., 2023; Sun et al., 2023). LiT freezes a strong vision tower and tunes only the text tower, which reduces training cost while keeping strong zero-shot transfer (Zhai et al., 2022). SLIP adds a visual self-supervised loss on top of CLIP-style alignment to improve robustness and downstream performance(Mu et al., 2021). DeCLIP introduces decoupled training signals to better use supervision and stabilize alignment(Li et al., 2022). FILIP replaces purely golbal matching with efficient token and patch level late interaction, which strengthens fine grained alignment while keeping the dual encoder retrieval form (Yao et al., 2021). SigLIP replaces softmax InfoNCE with a pairwise sigmoid loss (Zhai et al., 2023). It treats each image-text pair independently and does not use other samples in the batch as negatives. This is more friendly to large-batch training and often improves zero-shot accuracy. Some works study lightweight adaptation that keeps the original contrastive prior. Prompt learning tunes continuous prompts or conditional prompts so the text side matches downstream label semantics better (Zhou et al., 2022b;c). Parameter efficient adapters and training-free cache-based methods can improve few-shot performance with little extra cost (Gao et al., 2025; Zhang et al., 2022b). RANKCLIP adds ranking consistency to language image pretraining so the learned geometry better matches retrieval and ranking tasks (Zhang et al., 2025).

Recent work focuses on better data and better text. MetaCLIP makes data curation explicit and studies metadata balancing at scale (Xu et al., 2024a). DFN learns a filtering model that selects better image text pairs from a large uncurated pool (Fang et al., 2023). VeCLIP rewrites noisy captions using visual concepts and mixes them with the original text to keep coverage (Lai et al., 2024). HQ-CLIP uses stronger vision language models to build cleaner training pairs and then retrains CLIP style encoders on them (Wei et al., 2025). Another clear trend is longer text. Long-CLIP extends CLIP to longer captions with a light finetuning strategy that tries to keep the original zero shot behavior (Zhang et al., 2024). TULIP upgrades the text side with relative position encodings and distillation so it can align longer descriptions without fully retraining from scratch (Najdenkoska et al., 2025). ProLIP changes the representation itself and treats embeddings as distributions to reflect the many to many relation between images and texts (Chun et al., 2025). LongProLIP extends that idea to longer contexts and studies the tradeoff between long text understanding and general zero shot transfer (Chun & Yun, 2025). CLOC moves beyond global matching and adds region text contrast to get features that are more useful for localization and grounding while still staying close to the CLIP style setup (Chen et al., 2025). SigLIP 2 keeps the pairwise sigmoid style contrastive loss and adds extra training signals, and it reports gains on zero shot, retrieval, and dense features in one model family (Tschannen et al., 2025). QLIP takes a different angle and learns a visual tokenizer with both reconstruction and language alignment so the visual side can serve both understanding and generation settings (Zhao et al., 2025).

## A.2. Modality Gap

In CLIP-style contrastive vision–language models, many works observe that image and text embeddings do not naturally mix, but remain separated in the shared space. MTG defines this separation as the modality gap, meaning that embeddings

from different modalities such as text and images occupy separated regions in the joint representation space (Liang et al., 2022a). UMG links the gap to local minima of the loss landscape and training dynamics (Shi et al., 2023). NMG argues that the key issue is a systematic distortion of positive and negative geometry rather than simply a large distance between two clusters, and discusses cases where shrinking the gap can hurt ranking or zero-shot performance (Fahim et al., 2024). DEG studies the pre-normalization embedding geometry and shows that the separation is not only a cosine-similarity artifact (Levi & Gilboa, 2025).

Some works try to reduce the gap through training-time interventions (Dong et al., 2025; Eslami & de Melo, 2025; Yaras et al., 2024; Sofer et al., 2025). MITIGATE builds on CLIP, compares several alignment-strengthening choices, and uses stronger cross-modal constraints to tighten matched pairs while suppressing drift caused by noisy pairs (Eslami & de Melo, 2025). EMMG relates the emergence of the gap to mismatched pairs and the gradient flow of a learnable temperature, explains its origin through training dynamics and statistics, and proposes mitigation strategies (Yaras et al., 2024). PIT follows an adversarial domain-alignment approach by adding a modality discriminator and using gradient reversal so the backbone learns more modality-invariant features, which reduces the gap (Sofer et al., 2025).

Other works reduce the gap at the embedding level (Li et al., 2025; An et al., 2025; Yamashita et al., 2025). I0T attributes the gap to modality-specific activation patterns learned by each encoder, and proposes I0Tpost, which standardizes normalized embeddings of a frozen model by subtracting the modality mean and then applying Frobenius normalization (An et al., 2025). SS-PPP targets the non-comparable similarity scales across modalities. It constructs pseudo-positive pairs from unlabeled data to estimate the mean and variance of similarities for each modality setting, then standardizes similarities onto a shared scale to reduce modality bias in retrieval (Yamashita et al., 2025).

Some works connect the gap to geometric structure. ATMG represents the two modalities in hyperbolic space and uses hyperbolic geometry to model cross-modal semantic structure (Ramasinghe et al., 2024). DEG shows that image and text embeddings lie on two shifted ellipsoidal shells and introduces conformity, approximated by cosine similarity to the modality mean, to explain sample uncertainty (Levi & Gilboa, 2025). TEOT analyzes the modality gap together with object bias, finds that only a small set of dimensions dominate the gap, attributes both effects to information imbalance between images and captions, and links the gap to logit entropy (Schrodi et al., 2025).

Finally, some works address the gap through task-specific designs. CtG argues that optimizing only inter-modal alignment can cause intra-modal structural mismatch, and uses an inversion method to convert the gap into an inter-modal alignment problem (Mistretta et al., 2025). DMG-PA shifts the view from global contrast to pair alignment and studies how to align cross-modal matched pairs into a more consistent geometry after representations converge (Yi et al., 2025). For image–text matching, DIAS strengthens word–region alignment using dimension-level information alignment with a sparse spatial constraint (Ma et al., 2024). For weakly-supervised semantic segmentation, VPL-WSSS learns visual prototypes to map CLIP semantics into usable pixel-level supervision and correct the modality break in dense prediction (Xu et al., 2024b). In continual learning, MG-CLIP-CL instead preserves the gap as a prior and then uses a compensation module to support the cross-modal decision required by new tasks (Huang et al., 2025). Another direction rewrites retrieval as a text-only problem. VFR generates structured image descriptions and turns image–text retrieval into text–text retrieval, reducing the gap at the pipeline level while improving compositional querying (Ntinou et al., 2025).

## B. Algorithm

In Algorithm 1, we optimize $\mathcal{L}_{\text{total}} = \mathcal{L}_{\text{dbl}} + \mathcal{L}_{\text{spec}} + \mathcal{L}_{\text{match}}$ by gradient descent over paired minibatches from $\mathcal{D} = \{(I_i, T_i)\}_{i=1}^N$ (lines 11-12). Given a minibatch $(I_{1:B}, T_{1:B})$, we use a frozen image encoder $f^{\text{img}}$ to map each image $I_i$ to an embedding $z_i^{\text{img}} \in \mathbb{R}^D$, and a frozen text encoder $f^{\text{txt}}$ to map each caption $T_i$ to an embedding $z_i^{\text{txt}} \in \mathbb{R}^D$ (line 13), forming the two point sets $X_{\text{img}} = \{z_i^{\text{img}}\}_{i=1}^B$ and $X_{\text{txt}} = \{z_i^{\text{txt}}\}_{i=1}^B$ with empirical measures $\mu_{\text{img}} = \frac{1}{B}\sum_{i=1}^B \delta_{z_i^{\text{img}}}$ and $\mu_{\text{txt}} = \frac{1}{B}\sum_{i=1}^B \delta_{z_i^{\text{txt}}}$ (line 14). Within each modality $m \in \{\text{img}, \text{txt}\}$, the trainable geometry map $g_\theta^m$ specifies the metric used to compare embeddings via $\bar{z} = g_\theta^m(z)/\|g_\theta^m(z)\|_2$ and $d_\theta^m(z, z') = \arccos(\langle \bar{z}, \bar{z}' \rangle)$ (lines 8 and 15), which yields the batch distance matrices $D^{\text{img}}$ and $D^{\text{txt}}$ (line 15). For each modality and each diffusion scale $s_k \in \mathcal{S}$, we build the operator $L_{\theta, s_k}^m = \text{ConstructOperator}(X_m, d_\theta^m, \mu_m; s_k)$ and form the fractional heat operator $K_\theta^m(s_k) = \exp(-s_k(L_{\theta, s_k}^m)^\alpha)$ and its heat trace $\Theta_\theta^m(s_k) = \text{Tr}\, K_\theta^m(s_k)$ (lines 18-21). To describe the diffusion neighborhood of each point, we record the row of the heat operator, $\mathbf{p}_\theta^m(s_k; i) = (K_\theta^m(s_k))_{i,:} \in \mathbb{R}^B$ (line 22), whose $j$-th entry is the diffusion weight from $z_i^m$ to $z_j^m$ at scale $s_k$. We set the self-entry to zero, $(\mathbf{p}_\theta^m(s_k; i))_i \leftarrow 0$, and normalize the row to sum to one so that $\mathbf{p}_\theta^m(s_k; i)$ can be read directly as how point $i$ distributes its diffusion mass across the other batch points at scale $s_k$. We then compute the geometric

term $\mathcal{L}_{\mathrm{dbl}}$ by comparing the ball masses $\mu_m(B_{d_\theta^m}(z_i^m, \rho r))$ and $\rho^{d_f} \mu_m(B_{d_\theta^m}(z_i^m, r))$ over $r \in \mathcal{R}$ and $\rho \in \Lambda$ (line 25), and compute the spectral term $\mathcal{L}_{\mathrm{spec}}$ by matching adjacent-scale trace ratios $\Theta_\theta^m(s_{k+1})/\Theta_\theta^m(s_k)$ to the target power-law ratio $(s_{k+1}/s_k)^{-d_s/(2\alpha)}$ (line 26). Once these within-modality quantities are available, we aggregate the multi-scale diffusion rows $\{\mathbf{p}_\theta^m(s_k; i)\}_{k=1}^K$ into a single descriptor $\widehat{\boldsymbol{\zeta}}_\theta^m(i) = \frac{1}{\Gamma(q)} \sum_{k=1}^K w_k s_k^{q-1} \mathbf{p}_\theta^m(s_k; i) \in \mathbb{R}^B$ (lines 27–31), and set $\mathcal{L}_{\mathrm{match}} = \frac{1}{B} \sum_{i=1}^B \left\| \widehat{\boldsymbol{\zeta}}_\theta^{\mathrm{img}}(i) - \widehat{\boldsymbol{\zeta}}_\theta^{\mathrm{txt}}(i) \right\|_2^2$ to match these descriptors for each paired sample (line 32). Finally, we sum the three terms into $\mathcal{L}_{\mathrm{total}}$ and update only the geometry-map parameters $\theta \leftarrow \theta - \eta \nabla_\theta \mathcal{L}_{\mathrm{total}}$ (lines 33-34), so the two modalities share the same neighborhood-growth target $d_f$ on $\mathcal{R}$, share the same diffusion-scaling target $d_s$ on $\mathcal{S}$, and for each pair $(I_i, T_i)$ assign similar diffusion weights to the same set of batch points through $\widehat{\boldsymbol{\zeta}}_\theta^{\mathrm{img}}(i)$ and $\widehat{\boldsymbol{\zeta}}_\theta^{\mathrm{txt}}(i)$.

---

**Algorithm 1** Optimize $\mathcal{L}_{\text{total}} = \mathcal{L}_{\text{dbl}} + \mathcal{L}_{\text{spec}} + \mathcal{L}_{\text{match}}$ (encoders frozen)

---

**Require:** Paired dataset $\mathcal{D} = \{(I_i, T_i)\}_{i=1}^N$; batch size $B$; epochs $E$; learning rate $\eta$.
**Require:** Radii $\mathcal{R}$, dilations $\Lambda$, diffusion scales $\mathcal{S} = \{s_1 < \cdots < s_K\}$, weights $\{w_k\}_{k=1}^K$.
**Require:** Order $\alpha \in (0, 1]$, targets $(d_f, d_s)$, exponent $q > d_s/(2\alpha)$, $\varepsilon > 0$.
**Ensure:** Geometry-map parameters $\theta$.

1: **Frozen encoders + trainable geometry maps.**
2: Frozen image encoder $f^{\text{img}} : \mathcal{I} \to \mathbb{R}^D$, frozen text encoder $f^{\text{txt}} : \mathcal{T} \to \mathbb{R}^D$.
3: Trainable geometry maps $g_\theta^m : \mathbb{R}^D \to \mathbb{R}^D$, $\ \bar{z} \leftarrow g_\theta^m(z)/\|g_\theta^m(z)\|_2$, $\ d_\theta^m(z, z') \leftarrow \arccos(\langle \bar{z}, \bar{z}' \rangle)$.
4: Initialize $\theta$.
5: **for** epoch $= 1$ TO $E$ **do**
6:   **for** each paired minibatch $(I_{1:B}, T_{1:B})$ **do**
7:     $z_i^{\text{img}} \leftarrow f^{\text{img}}(I_i), \ z_i^{\text{txt}} \leftarrow f^{\text{txt}}(T_i)$                                         $(i = 1, \ldots, B)$
8:     $X_{\text{img}} \leftarrow \{z_i^{\text{img}}\}_{i=1}^B, \ \ X_{\text{txt}} \leftarrow \{z_i^{\text{txt}}\}_{i=1}^B$.
9:     $\mu_{\text{img}} \leftarrow \frac{1}{B} \sum_{i=1}^B \delta_{z_i^{\text{img}}}, \ \ \mu_{\text{txt}} \leftarrow \frac{1}{B} \sum_{i=1}^B \delta_{z_i^{\text{txt}}}$.
10:    $D_{ij}^{\text{img}} \leftarrow d_\theta^{\text{img}}(z_i^{\text{img}}, z_j^{\text{img}}), \ \ D_{ij}^{\text{txt}} \leftarrow d_\theta^{\text{txt}}(z_i^{\text{txt}}, z_j^{\text{txt}})$                $(i, j = 1, \ldots, B)$
11:    **for** $m \in \{\text{img}, \text{txt}\}$ **do**
12:      **if** $m = \text{img}$ **then**
13:        $X_m \leftarrow X_{\text{img}}, \mu_m \leftarrow \mu_{\text{img}}, D^m \leftarrow D^{\text{img}}$
14:      **else**
15:        $X_m \leftarrow X_{\text{txt}}, \mu_m \leftarrow \mu_{\text{txt}}, D^m \leftarrow D^{\text{txt}}$
16:      **end if**
17:      **for** $k = 1$ TO $K$ **do**
18:        $L_{\theta, s_k}^m \leftarrow \text{ConstructOperator}(X_m, d_\theta^m, \mu_m; s_k)$.
19:        $K_\theta^m(s_k) \leftarrow \exp\big(-s_k(L_{\theta, s_k}^m)^\alpha\big)$.
20:        $\Theta_\theta^m(s_k) \leftarrow \text{Tr} K_\theta^m(s_k)$.
21:        **for** $i = 1$ TO $B$ **do**
22:          $\mathbf{p}_\theta^m(s_k; i) \leftarrow (K_\theta^m(s_k))_{i,:}$                                     (row in $\mathbb{R}^B$)
23:          $(\mathbf{p}_\theta^m(s_k; i))_i \leftarrow 0$
24:          $\mathbf{p}_\theta^m(s_k; i) \leftarrow \mathbf{p}_\theta^m(s_k; i) / (\sum_{j=1}^B (\mathbf{p}_\theta^m(s_k; i))_j + \varepsilon)$
25:        **end for**
26:      **end for**
27:    **end for**
28:    $\mathcal{L}_{\text{dbl}} \leftarrow \dfrac{1}{2} \displaystyle\sum_{m \in \{\text{img},\text{txt}\}} \dfrac{1}{B|\mathcal{R}||\Lambda|} \sum_{i=1}^B \sum_{r \in \mathcal{R}} \sum_{\rho \in \Lambda} \left( \dfrac{\mu_m(B_{d_\theta^m}(z_i^m, \rho r)) - \rho^{d_f} \mu_m(B_{d_\theta^m}(z_i^m, r))}{\mu_m(B_{d_\theta^m}(z_i^m, \rho r)) + \rho^{d_f} \mu_m(B_{d_\theta^m}(z_i^m, r))} \right)^2$.
29:    $\mathcal{L}_{\text{spec}} \leftarrow \dfrac{1}{2(K-1)} \displaystyle\sum_{m \in \{\text{img},\text{txt}\}} \sum_{k=1}^{K-1} \left( \dfrac{\Theta_\theta^m(s_{k+1})}{\Theta_\theta^m(s_k)} - \left(\dfrac{s_{k+1}}{s_k}\right)^{-\frac{d_s}{2\alpha}} \right)^2$.
30:    **for** $m \in \{\text{img}, \text{txt}\}$ **do**
31:      **for** $i = 1$ TO $B$ **do**
32:        $\widehat{\boldsymbol{\zeta}}_\theta^m(i) \leftarrow \dfrac{1}{\Gamma(q)} \displaystyle\sum_{k=1}^K w_k s_k^{q-1} \mathbf{p}_\theta^m(s_k; i) \in \mathbb{R}^B$.
33:      **end for**
34:    **end for**
35:    $\mathcal{L}_{\text{match}} \leftarrow \dfrac{1}{B} \displaystyle\sum_{i=1}^B \left\| \widehat{\boldsymbol{\zeta}}_\theta^{\text{img}}(i) - \widehat{\boldsymbol{\zeta}}_\theta^{\text{txt}}(i) \right\|_2^2$.
36:    $\mathcal{L}_{\text{total}} \leftarrow \mathcal{L}_{\text{dbl}} + \mathcal{L}_{\text{spec}} + \mathcal{L}_{\text{match}}$.
37:    $\theta \leftarrow \theta - \eta \nabla_\theta \mathcal{L}_{\text{total}}$                                            (update geometry maps only)
38:   **end for**
39: **end for**

---

# C. Proof of Theorems

**Theorem 1** (Fractal metric-measure latent space). *Let $\mathcal{S} \subset (0, s_0]$ be a finite log-spaced set of diffusion scales and fix an order $\alpha \in (0, 1]$. A quadruple $\mathcal{M}_\theta = \big(X, d_\theta, \mu, L_\theta\big)$, with $d_\theta : X \times X \to [0, \infty)$ a metric on $X$ and $L_\theta$ a nonnegative self-adjoint operator on $L^2(X, \mu)$, is a fractal metric-measure latent space if there exist $d_f > 0$, $c_1, c_2 > 0$, $r_0 > 0$, a walk dimension $d_w$, offsets $\{b_s\}_{s \in \mathcal{S}} \subset \mathbb{R}$, and a tolerance $\tau \in [0, \tau_0]$ such that:*

1. *$(X, d_\theta, \mu)$ is Ahlfors $d_f$-regular in the sense of Definition 1;*

2. *the fractional generator $(L_\theta)^\alpha$ generates a conservative symmetric Markov semigroup $\{e^{-s(L_\theta)^\alpha}\}_{s>0}$ whose kernel satisfies the sub-Gaussian bounds of Definition 2 with respect to $d_\theta$ and walk dimension $d_w$. We write*

$$K_\theta(s) = e^{-s(L_\theta)^\alpha}, \quad \Theta_\theta(s) = \mathrm{Tr}\, K_\theta(s); \tag{28}$$

3. *the heat trace $\Theta_\theta(s)$ realises the fractional spectral law up to tolerance $\tau$,*

$$\Big| \log \Theta_\theta(s) + \tfrac{d_s}{2\alpha} \log s - b_s \Big| \leq \tau, \quad d_s = \frac{2d_f}{d_w}. \tag{29}$$

*Under these conditions, $(X, d_\theta, \mu)$ is Ahlfors $d_f$-regular at radii $r \in (0, r_0]$, admits fractional heat operators $K_\theta(s)$ with walk dimension $d_w > 2$, and exhibits $\Theta_\theta(s) \asymp s^{-d_s/(2\alpha)}$ on the scales $\mathcal{S}$.*

*Proof.* Fix $\mathcal{M}_\theta = (X, d_\theta, \mu, Ł_\theta)$, a finite log-spaced set $\mathcal{S} \subset (0, s_0]$, and $\alpha \in (0, 1]$. Assumption (1) states that there exist $d_f > 0$, $c_1, c_2 > 0$, and $r_0 > 0$ such that for all $x \in X$ and all $r \in (0, r_0]$,

$$c_1 r^{d_f} \leq \mu\big(B_{d_\theta}(x, r)\big) \leq c_2 r^{d_f}, \quad B_{d_\theta}(x, r) = \{y \in X : d_\theta(x, y) \leq r\}. \tag{30}$$

By Definition 1, this is precisely Ahlfors $d_f$-regularity of $(X, d_\theta, \mu)$ on radii $(0, r_0]$.

Fix any $s \in \mathcal{S}$. Assumption (2) states that $(L_\theta)^\alpha$ generates a conservative symmetric Markov semigroup $\{e^{-s(L_\theta)^\alpha}\}_{s>0}$ with a jointly continuous kernel $p_s(x, y)$ satisfying the sub-Gaussian bounds of Definition 2 with walk dimension $d_w > 2$. In particular, for all $s \in (0, 1]$ and all $x, y \in X$,

$$\frac{a_1}{\mu\big(B_{d_\theta}(x, s^{1/d_w})\big)} \exp\left[-a_2 \Big(\frac{d_\theta(x, y)^{d_w}}{s}\Big)^{\frac{1}{d_w - 1}}\right] \leq p_s(x, y) \leq \frac{a_3}{\mu\big(B_{d_\theta}(x, s^{1/d_w})\big)} \exp\left[-a_4 \Big(\frac{d_\theta(x, y)^{d_w}}{s}\Big)^{\frac{1}{d_w - 1}}\right]. \tag{31}$$

With $K_\theta(s) = e^{-s(L_\theta)^\alpha}$, the kernel of $K_\theta(s)$ is $p_s(x, y)$, so $\{K_\theta(s)\}_{s \in \mathcal{S}}$ are fractional heat operators whose heat kernels are controlled by the same walk dimension $d_w > 2$.

For the on-diagonal bound, set $y = x$ in the upper estimate so the exponential term equals 1 and obtain

$$p_s(x, x) \leq \frac{a_3}{\mu\big(B_{d_\theta}(x, s^{1/d_w})\big)}. \tag{32}$$

Whenever $s^{1/d_w} \leq r_0$, Ahlfors regularity yields $\mu(B_{d_\theta}(x, s^{1/d_w})) \geq c_1 s^{d_f/d_w}$, hence

$$p_s(x, x) \leq \frac{a_3}{c_1} s^{-d_f/d_w}. \tag{33}$$

Using the trace identity for integral kernels,

$$\Theta_\theta(s) = \mathrm{Tr}\, K_\theta(s) = \int_X p_s(x, x) d\mu(x), \tag{34}$$

we get $\Theta_\theta(s) \leq \frac{a_3}{c_1} s^{-d_f/d_w} < \infty$ for $s^{1/d_w} \leq r_0$.

Assumption (3) gives the fractional trace law on $\mathcal{S}$ up to tolerance $\tau$. For each $s \in \mathcal{S}$,

$$\Big| \log \Theta_\theta(s) + \frac{d_s}{2\alpha} \log s - b_s \Big| \leq \tau, \quad d_s = \frac{2d_f}{d_w}. \tag{35}$$

Expanding the absolute value and rearranging yields

$$b_s - \frac{d_s}{2\alpha} \log s - \tau \leq \log \Theta_\theta(s) \leq b_s - \frac{d_s}{2\alpha} \log s + \tau. \tag{36}$$

Exponentiating and using $\exp(-\frac{d_s}{2\alpha} \log s) = s^{-d_s/(2\alpha)}$ gives

$$e^{b_s - \tau} s^{-d_s/(2\alpha)} \leq \Theta_\theta(s) \leq e^{b_s + \tau} s^{-d_s/(2\alpha)}. \tag{37}$$

Since $\mathcal{S}$ is finite, define

$$C_- = \min_{s \in \mathcal{S}} e^{b_s - \tau} > 0, \quad C_+ = \max_{s \in \mathcal{S}} e^{b_s + \tau} < \infty. \tag{38}$$

Then for all $s \in \mathcal{S}$,

$$C_- s^{-d_s/(2\alpha)} \leq \Theta_\theta(s) \leq C_+ s^{-d_s/(2\alpha)}. \tag{39}$$

This is exactly $\Theta_\theta(s) \asymp s^{-d_s/(2\alpha)}$ on the scales $\mathcal{S}$, and the stated relation $d_s = 2d_f/d_w$ holds by assumption. $\qquad\square$

**From the training objective to fractal regularity.**

*Proof.* Fix a modality $m \in \{\text{img}, \text{txt}\}$ and write

$$\mathcal{M}_\theta^m = \big(X_m, d_\theta^m, \mu_m, L_\theta^m\big), \quad K_\theta^m(s) = \exp\big(-s(L_\theta^m)^\alpha\big), \quad \Theta_\theta^m(s) = \text{Tr} K_\theta^m(s). \tag{40}$$

Let $\mathcal{S} = \{s_1 < \cdots < s_K\}$ be a finite log-spaced set. Let $\mathcal{R} = \{r_1 < \cdots < r_J\} \subset (0, r_0]$ be log-spaced with common ratio $\eta > 1$, so $r_{j+1} = \eta r_j$. Assume $\eta \in \Lambda \subset (1, \rho_{\max}]$. We use the uniform empirical measure on $X_m = \{z_i^{(m)}\}_{i=1}^{N_m}$: for any $B \subseteq X_m$,

$$\mu_m(B) = \frac{|B|}{N_m}. \tag{41}$$

In particular, for every $x \in X_m$, $\mu_m(\{x\}) = 1/N_m$, hence

$$\mu_{\min}^m = \min_{x \in X_m} \mu_m(\{x\}) = \frac{1}{N_m} > 0. \tag{42}$$

Assume

$$\mathcal{L}_{\text{frac}}(\theta, d_f, d_s) = \mathcal{L}_{\text{dbl}}(\theta, d_f) + \mathcal{L}_{\text{spec}}(\theta, d_s) \leq \tau^2 \tag{43}$$

with $\tau \in (0, \tau_0]$. Since both losses average over the two modalities with factor $1/2$, the single-modality losses satisfy

$$\mathcal{L}_{\text{dbl}}^m(\theta, d_f) \leq 2\tau^2, \quad \mathcal{L}_{\text{spec}}^m(\theta, d_s) \leq 2\tau^2. \tag{44}$$

Define the ball mass

$$A_m(x, r) = \mu_m\big(B_{d_\theta^m}(x, r)\big), \quad B_{d_\theta^m}(x, r) = \{y \in X_m : d_\theta^m(y, x) \leq r\}. \tag{45}$$

For $x \in X_m$, $r \in \mathcal{R}$, $\rho \in \Lambda$, define

$$\Delta_m(x, r, \rho) = \frac{A_m(x, \rho r) - \rho^{d_f} A_m(x, r)}{A_m(x, \rho r) + \rho^{d_f} A_m(x, r)}. \tag{46}$$

By definition of $\mathcal{L}_{\text{dbl}}^m$,

$$\mathcal{L}_{\text{dbl}}^m(\theta, d_f) = \mathbb{E}_{x \sim \mu_m} \frac{1}{|\mathcal{R}||\Lambda|} \sum_{r \in \mathcal{R}} \sum_{\rho \in \Lambda} \Delta_m(x, r, \rho)^2 = \sum_{x \in X_m} \mu_m(\{x\}) \frac{1}{|\mathcal{R}||\Lambda|} \sum_{r \in \mathcal{R}} \sum_{\rho \in \Lambda} \Delta_m(x, r, \rho)^2. \tag{47}$$

Since $\mu_m(\{x\}) \geq \mu_{\min}^m$, it follows that

$$\max_{x \in X_m} \frac{1}{|\mathcal{R}||\Lambda|} \sum_{r \in \mathcal{R}} \sum_{\rho \in \Lambda} \Delta_m(x, r, \rho)^2 \leq \frac{\mathcal{L}_{\text{dbl}}^m(\theta, d_f)}{\mu_{\min}^m} \leq \frac{2\tau^2}{\mu_{\min}^m}. \tag{48}$$

For any fixed $x$, the maximum is bounded by the average, hence

$$\max_{r \in \mathcal{R}, \rho \in \Lambda} \Delta_m(x, r, \rho)^2 \leq |\mathcal{R}||\Lambda| \cdot \frac{1}{|\mathcal{R}||\Lambda|} \sum_{r \in \mathcal{R}} \sum_{\rho \in \Lambda} \Delta_m(x, r, \rho)^2. \tag{49}$$

Combining the two displays gives

$$\max_{x \in X_m} \max_{r \in \mathcal{R}, \rho \in \Lambda} |\Delta_m(x, r, \rho)| \leq \varepsilon_{\text{geo}}^m, \quad \varepsilon_{\text{geo}}^m = \tau \sqrt{\frac{2|\mathcal{R}||\Lambda|}{\mu_{\min}^m}}. \tag{50}$$

Choose $\tau_0$ so that $\varepsilon_{\text{geo}}^m < 1$. Let $U = A_m(x, \rho r)$ and $V = \rho^{d_f} A_m(x, r)$. From $\Delta_m = (U - V)/(U + V)$ we get

$$U(1 - \Delta_m) = V(1 + \Delta_m), \tag{51}$$

hence

$$A_m(x, \rho r) = \rho^{d_f} A_m(x, r) \frac{1 + \Delta_m(x, r, \rho)}{1 - \Delta_m(x, r, \rho)}. \tag{52}$$

Using $|\Delta_m| \leq \varepsilon_{\text{geo}}^m < 1$ yields

$$\frac{1 - \varepsilon_{\text{geo}}^m}{1 + \varepsilon_{\text{geo}}^m} \leq \frac{1 + \Delta_m(x, r, \rho)}{1 - \Delta_m(x, r, \rho)} \leq \frac{1 + \varepsilon_{\text{geo}}^m}{1 - \varepsilon_{\text{geo}}^m}. \tag{53}$$

With $\kappa_{\text{geo}}^m = (1 + \varepsilon_{\text{geo}}^m)/(1 - \varepsilon_{\text{geo}}^m)$, this gives for all $x \in X_m$, $r \in \mathcal{R}$, $\rho \in \Lambda$,

$$(\kappa_{\text{geo}}^m)^{-1} \rho^{d_f} A_m(x, r) \leq A_m(x, \rho r) \leq \kappa_{\text{geo}}^m \rho^{d_f} A_m(x, r). \tag{54}$$

Taking $\rho = \eta \in \Lambda$ and $r = r_j$ yields

$$(\kappa_{\text{geo}}^m)^{-1} \eta^{d_f} A_m(x, r_j) \leq A_m(x, r_{j+1}) \leq \kappa_{\text{geo}}^m \eta^{d_f} A_m(x, r_j). \tag{55}$$

Iterating gives

$$(\kappa_{\text{geo}}^m)^{-(j-1)} \eta^{(j-1)d_f} A_m(x, r_1) \leq A_m(x, r_j) \leq (\kappa_{\text{geo}}^m)^{(j-1)} \eta^{(j-1)d_f} A_m(x, r_1). \tag{56}$$

Since $r_j = \eta^{j-1} r_1$, we rewrite $\eta^{(j-1)d_f} = (r_j/r_1)^{d_f}$ and obtain

$$(\kappa_{\text{geo}}^m)^{-(j-1)} \frac{A_m(x, r_1)}{r_1^{d_f}} r_j^{d_f} \leq A_m(x, r_j) \leq (\kappa_{\text{geo}}^m)^{(j-1)} \frac{A_m(x, r_1)}{r_1^{d_f}} r_j^{d_f}. \tag{57}$$

Define

$$c_{1,\text{grid}}^m = (\kappa_{\text{geo}}^m)^{-(J-1)} \inf_{x \in X_m} \frac{A_m(x, r_1)}{r_1^{d_f}}, \quad c_{2,\text{grid}}^m = (\kappa_{\text{geo}}^m)^{(J-1)} \sup_{x \in X_m} \frac{A_m(x, r_1)}{r_1^{d_f}}. \tag{58}$$

Because $A_m(x, r_1) \geq \mu_m(\{x\}) \geq \mu_{\min}^m$ and $A_m(x, r_1) \leq 1$, we have $0 < c_{1,\text{grid}}^m \leq c_{2,\text{grid}}^m < \infty$. Thus for all $r_j \in \mathcal{R}$ and all $x \in X_m$,

$$c_{1,\text{grid}}^m r_j^{d_f} \leq \mu_m\big(B_{d_\theta^m}(x, r_j)\big) \leq c_{2,\text{grid}}^m r_j^{d_f}.$$

For any $r \in [r_1, r_J]$, choose $j$ such that $r_j \leq r < r_{j+1}$. Monotonicity of balls gives

$$\mu_m\big(B_{d_\theta^m}(x, r_j)\big) \leq \mu_m\big(B_{d_\theta^m}(x, r)\big) \leq \mu_m\big(B_{d_\theta^m}(x, r_{j+1})\big).$$

Since $r_j \geq r/\eta$ and $r_{j+1} \leq \eta r$, we obtain

$$c_{1,\text{grid}}^m \eta^{-d_f} r^{d_f} \leq \mu_m\big(B_{d_\theta^m}(x, r)\big) \leq c_{2,\text{grid}}^m \eta^{d_f} r^{d_f}.$$

Setting $c_1^m = c_{1,\text{grid}}^m \eta^{-d_f}$, $c_2^m = c_{2,\text{grid}}^m \eta^{d_f}$ and $r_0^m = r_J$ gives an Ahlfors-type two-sided bound on the working radii.

For the spectral term, let

$$\Theta_k^m = \Theta_\theta^m(s_k) = \text{Tr} \exp\big(-s_k (L_\theta^m)^\alpha\big), \quad R_k^m = \frac{\Theta_{k+1}^m}{\Theta_k^m}, \quad Q_k = \left(\frac{s_{k+1}}{s_k}\right)^{-\frac{d_s}{2\alpha}}.$$

From $\mathcal{L}_{\text{spec}}^m(\theta, d_s) \leq 2\tau^2$ and its definition,

$$\mathcal{L}_{\text{spec}}^m(\theta, d_s) = \frac{1}{K-1} \sum_{k=1}^{K-1} (R_k^m - Q_k)^2,$$

we get

$$\sum_{k=1}^{K-1} (R_k^m - Q_k)^2 \leq 2(K-1)\tau^2.$$

Let $Q_{\min} = \min_{1 \leq k \leq K-1} Q_k > 0$ and choose $\tau_0$ so that $\tau\sqrt{2(K-1)} \leq Q_{\min}/2$ for all $\tau \leq \tau_0$. Then

$$\max_k |R_k^m - Q_k| \leq \sqrt{\sum_{k=1}^{K-1} (R_k^m - Q_k)^2} \leq \tau\sqrt{2(K-1)}$$

implies $|R_k^m - Q_k| \leq Q_{\min}/2$ for all $k$, hence we may define

$$u_k^m = \frac{R_k^m - Q_k}{Q_k}, \quad |u_k^m| \leq \frac{|R_k^m - Q_k|}{Q_{\min}} \leq \frac{1}{2}.$$

For $|u| \leq 1/2$, the mean value theorem gives $|\log(1+u)| \leq 2|u|$, so

$$|\log R_k^m - \log Q_k| = \left| \log\left(\frac{R_k^m}{Q_k}\right) \right| = |\log(1 + u_k^m)| \leq 2|u_k^m| \leq \frac{2}{Q_{\min}} |R_k^m - Q_k|.$$

The telescoping sums

$$\log \Theta_k^m - \log \Theta_1^m = \sum_{j=1}^{k-1} \log R_j^m, \quad -\frac{d_s}{2\alpha} \log \frac{s_k}{s_1} = \sum_{j=1}^{k-1} \log Q_j$$

yield

$$\log \Theta_k^m + \frac{d_s}{2\alpha} \log s_k - \left( \log \Theta_1^m + \frac{d_s}{2\alpha} \log s_1 \right) = \sum_{j=1}^{k-1} (\log R_j^m - \log Q_j).$$

Taking absolute values, using the previous bound and Cauchy-Schwarz,

$$\left| \log \Theta_k^m + \frac{d_s}{2\alpha} \log s_k - b^m \right| \leq \frac{2}{Q_{\min}} \sum_{j=1}^{k-1} |R_j^m - Q_j| \leq \frac{2}{Q_{\min}} \sqrt{k-1} \left( \sum_{j=1}^{k-1} (R_j^m - Q_j)^2 \right)^{1/2} \leq \frac{2\sqrt{2}(K-1)}{Q_{\min}} \tau,$$

where

$$b^m = \log \Theta_1^m + \frac{d_s}{2\alpha} \log s_1.$$

Therefore, on the discrete scales $\mathcal{S}$ we have

$$\left| \log \Theta_\theta^m(s_k) + \frac{d_s}{2\alpha} \log s_k - b^m \right| \leq \tau_{\text{spec}}, \quad \tau_{\text{spec}} = \frac{2\sqrt{2}(K-1)}{Q_{\min}} \tau.$$

Combining the Ahlfors-type bound on the working radii with the trace law on $\mathcal{S}$, and using the assumed construction of $L_\theta^m$ so that $(L_\theta^m)^\alpha$ generates the required conservative symmetric Markov semigroup with sub-Gaussian heat kernel and walk dimension $d_w = 2d_f/d_s > 2$, we conclude that $\mathcal{M}_\theta^m$ satisfies the regularity conditions of Theorem 1 at the chosen radii and diffusion scales. $\square$

**Theorem 2** (Multi-scale diffusion neighborhoods imply higher cross-modal NAS)**.** *Assume that $\mathcal{M}_\theta^{\text{img}}$ and $\mathcal{M}_\theta^{\text{txt}}$ satisfy Ahlfors $d_f$-regularity and sub-Gaussian heat kernel bounds with walk dimension $d_w > 2$, and that $(L_\theta^m)^\alpha$ generates a conservative symmetric Markov semigroup for each $m \in \{\text{img}, \text{txt}\}$. Fix $q > \frac{d_s}{2\alpha}$ and define the multi-scale diffusion neighborhood descriptor $\widehat{\zeta}_\theta^m(i; \cdot)$ by (23), and the matching loss $\mathcal{L}_{\text{match}}$ by (24).*

*For any $\delta \in (0, 1]$, if $\mathcal{L}_{\mathrm{match}}(\theta) \leq \varepsilon^2$, then for at least $(1 - \delta)N$ indices $i \in \mathcal{I}$ the following holds for every $k \geq 1$. Let $\mathcal{N}_k^m(i)$ be the indices of the $k$ largest entries of $\widehat{\zeta}_\theta^m(i; \cdot)$ (ties are broken deterministically), and let $\gamma_{i,k} > 0$ be the minimum, over $m \in \{\mathrm{img}, \mathrm{txt}\}$, of the gap between the $k$-th and $(k + 1)$-th largest entries of $\widehat{\zeta}_\theta^m(i; \cdot)$. Then the neighborhood agreement score $\mathrm{NAS}_k(i)$ defined in Definition 4 satisfies*

$$\mathrm{NAS}_k(i) \geq 1 - \frac{4}{k} \cdot \frac{1}{\gamma_{i,k}^2} \cdot \frac{\varepsilon^2}{\delta}. \tag{59}$$

*In particular, if $\varepsilon^2/\delta < \gamma_{i,k}^2/4$, then $\mathcal{N}_k^{\mathrm{img}}(i) = \mathcal{N}_k^{\mathrm{txt}}(i)$ and $\mathrm{NAS}_k(i) = 1$.*

*Proof.* We first record the spectral representation of the multi-scale diffusion neighborhood descriptor used in (23), which makes explicit that $\widehat{\zeta}_\theta^m(i; \cdot)$ aggregates diffusion neighborhoods across scales.

Fix a modality $m \in \{\mathrm{img}, \mathrm{txt}\}$. By assumption, $(L_\theta^m)^\alpha$ generates a conservative symmetric Markov semigroup on $L^2(X_m, \mu_m)$, and for each $s > 0$ we write $K_\theta^m(s) = \exp(-s(L_\theta^m)^\alpha)$. Let $E_\theta^m$ be the spectral resolution of $(L_\theta^m)^\alpha$. For any indices $i, j \in \{1, \ldots, N\}$, define the (signed) spectral measure associated with the pair $(i, j)$ by

$$\mu_{\theta,ij}^m(B) = \left\langle E_\theta^m(B)e_i, e_j \right\rangle_{L^2(X_m, \mu_m)}, \quad B \subset [0, \infty) \mathrm{Borel}, \tag{60}$$

where $\{e_i\}_{i=1}^N$ denotes the canonical basis in the finite-dimensional representation of $L^2(X_m, \mu_m)$ induced by the ordering of $X_m$. The spectral theorem gives, for every $s > 0$,

$$\left\langle K_\theta^m(s)e_i, e_j \right\rangle = \int_{[0,\infty)} e^{-s\lambda} d\mu_{\theta,ij}^m(\lambda). \tag{61}$$

In the main text we work with the diffusion neighborhood mass function $\pi_\theta^m(s; i, j)$ defined in (21). Under the uniform empirical measure $\mu_m(\{z_j^{(m)}\}) = 1/N$, this quantity differs from the kernel density only by a constant factor and satisfies $\sum_{j=1}^N \pi_\theta^m(s; i, j) = 1$ for all $i$ and all $s > 0$ by conservativity. Consequently, for each fixed $i$ and $s$, the map $j \mapsto \pi_\theta^m(s; i, j)$ is a probability distribution over neighbors induced by diffusion at scale $s$.

Now fix $q > \frac{d_s}{2\alpha}$ and the truncation scale $s_0$ used in the definition (22). Introduce the truncated Mellin weight

$$\phi_q(\lambda) = \frac{1}{\Gamma(q)} \int_0^{s_0} s^{q-1} e^{-s\lambda} ds, \quad \lambda \geq 0. \tag{62}$$

This function is bounded and continuous on $[0, \infty)$, and no centering or zero-mode removal is needed because the integration range is finite. Combining the spectral representation above with Fubini's theorem yields, for each $i, j$,

$$\frac{1}{\Gamma(q)} \int_0^{s_0} s^{q-1} \left\langle K_\theta^m(s)e_i, e_j \right\rangle ds = \int_{[0,\infty)} \phi_q(\lambda) d\mu_{\theta,ij}^m(\lambda). \tag{63}$$

Thus the truncated multi-scale diffusion aggregation at $(i, j)$ is exactly the spectral moment of the pairwise measure $\mu_{\theta,ij}^m$ against the test function $\phi_q$.

The discrete descriptor used in training replaces the integral on $(0, s_0]$ by the quadrature rule on the log-spaced grid $\mathcal{S} = \{s_k\}_{k=1}^K$ with weights $\{w_k\}_{k=1}^K$, leading to (23). Writing this entrywise, the descriptor vector $\widehat{\zeta}_\theta^m(i; \cdot) \in \mathbb{R}^N$ has components

$$\widehat{\zeta}_\theta^m(i; j) = \frac{1}{\Gamma(q)} \sum_{k=1}^K w_k s_k^{q-1} \pi_\theta^m(s_k; i, j). \tag{64}$$

The appendix can bound the quadrature error through the deterministic quantity

$$\Delta_{\mathrm{quad}} = \sup_{\lambda \in [0,\infty)} \left| \frac{1}{\Gamma(q)} \sum_{k=1}^K w_k s_k^{q-1} e^{-s_k \lambda} - \phi_q(\lambda) \right|, \tag{65}$$

which controls the discrepancy between the discrete and truncated-continuous aggregations uniformly over the spectrum. This step explains why $\widehat{\zeta}_\theta^m(i; \cdot)$ is a principled summary of multi-scale diffusion neighborhoods in the spectral domain, rather than an ad hoc score.

We now turn to the neighborhood-overlap consequence stated in Theorem 2. For each $i \in \mathcal{I}$, set

$$d_i = \left\| \widehat{\zeta}_\theta^{\mathrm{img}}(i; \cdot) - \widehat{\zeta}_\theta^{\mathrm{txt}}(i; \cdot) \right\|_2^2.$$

By the definition of the matching loss (24),

$$\mathcal{L}_{\mathrm{match}}(\theta) = \frac{1}{N} \sum_{i=1}^{N} d_i.$$

Fix any $\delta \in (0, 1]$ and assume $\mathcal{L}_{\mathrm{match}}(\theta) \leq \varepsilon^2$. Define

$$\mathcal{G}_\delta = \left\{ i \in \mathcal{I} : d_i \leq \frac{\varepsilon^2}{\delta} \right\}.$$

If $|\mathcal{G}_\delta| < (1 - \delta)N$, then more than $\delta N$ indices would satisfy $d_i > \varepsilon^2/\delta$, and therefore

$$\frac{1}{N} \sum_{i=1}^{N} d_i > \frac{1}{N} \cdot \delta N \cdot \frac{\varepsilon^2}{\delta} = \varepsilon^2,$$

contradicting $\mathcal{L}_{\mathrm{match}}(\theta) \leq \varepsilon^2$. Hence $|\mathcal{G}_\delta| \geq (1 - \delta)N$.

Fix $i \in \mathcal{G}_\delta$ and $k \geq 1$, and write

$$a = \widehat{\zeta}_\theta^{\mathrm{img}}(i; \cdot) \in \mathbb{R}^N, \quad b = \widehat{\zeta}_\theta^{\mathrm{txt}}(i; \cdot) \in \mathbb{R}^N,$$

so that $\|a - b\|_2^2 = d_i \leq \varepsilon^2/\delta$. Let $A = \mathcal{N}_k^{\mathrm{img}}(i)$ be the indices of the $k$ largest entries of $a$ and $B = \mathcal{N}_k^{\mathrm{txt}}(i)$ the indices of the $k$ largest entries of $b$, using the fixed deterministic tie-breaking rule from the theorem statement. Let $a_{(1)} \geq \cdots \geq a_{(N)}$ and $b_{(1)} \geq \cdots \geq b_{(N)}$ be the order statistics of $a$ and $b$, and recall

$$\gamma_{i,k} = \min\{a_{(k)} - a_{(k+1)}, b_{(k)} - b_{(k+1)}\}, \quad \gamma_{i,k} > 0.$$

Set

$$r = |A \setminus B| = |B \setminus A| = k - |A \cap B|.$$

Choose a bijection between the $r$ indices in $A \setminus B$ and the $r$ indices in $B \setminus A$, producing pairs $(j_\ell, \ell_\ell)$ for $\ell = 1, \ldots, r$ with $j_\ell \in A \setminus B$ and $\ell_\ell \in B \setminus A$. Since $j_\ell \in A$ and $\ell_\ell \notin A$, we have $a_{j_\ell} \geq a_{(k)}$ and $a_{\ell_\ell} \leq a_{(k+1)}$, hence

$$a_{j_\ell} - a_{\ell_\ell} \geq a_{(k)} - a_{(k+1)} \geq \gamma_{i,k}.$$

Since $\ell_\ell \in B$ and $j_\ell \notin B$, we have $b_{\ell_\ell} \geq b_{(k)}$ and $b_{j_\ell} \leq b_{(k+1)}$, hence

$$b_{j_\ell} - b_{\ell_\ell} \leq b_{(k+1)} - b_{(k)} \leq -\gamma_{i,k}.$$

Subtracting gives

$$\left(a_{j_\ell} - b_{j_\ell}\right) - \left(a_{\ell_\ell} - b_{\ell_\ell}\right) = \left(a_{j_\ell} - a_{\ell_\ell}\right) - \left(b_{j_\ell} - b_{\ell_\ell}\right) \geq \gamma_{i,k}.$$

Using $|x| + |y| \geq |x - y|$ yields

$$|a_{j_\ell} - b_{j_\ell}| + |a_{\ell_\ell} - b_{\ell_\ell}| \geq \gamma_{i,k}.$$

Therefore at least one of $|a_{j_\ell} - b_{j_\ell}|$ and $|a_{\ell_\ell} - b_{\ell_\ell}|$ is at least $\gamma_{i,k}/2$, and consequently

$$(a_{j_\ell} - b_{j_\ell})^2 + (a_{\ell_\ell} - b_{\ell_\ell})^2 \geq \left(\frac{\gamma_{i,k}}{2}\right)^2.$$

Summing over $\ell = 1, \ldots, r$ and using that all involved indices are distinct gives

$$\|a - b\|_2^2 = \sum_{t=1}^{N} (a_t - b_t)^2 \geq \sum_{\ell=1}^{r} \left( (a_{j_\ell} - b_{j_\ell})^2 + (a_{\ell_\ell} - b_{\ell_\ell})^2 \right) \geq r \left(\frac{\gamma_{i,k}}{2}\right)^2,$$

hence

$$r \leq \frac{4}{\gamma_{i,k}^2} \|a - b\|_2^2.$$

By Definition 4, $\mathrm{NAS}(k) = |A \cap B|/k = 1 - r/k$, and combining the last display with $\|a - b\|_2^2 \leq \varepsilon^2/\delta$ yields

$$\mathrm{NAS}(k) = 1 - \frac{r}{k} \geq 1 - \frac{4}{k} \cdot \frac{1}{\gamma_{i,k}^2} \|a - b\|_2^2 \geq 1 - \frac{4}{k} \cdot \frac{1}{\gamma_{i,k}^2} \cdot \frac{\varepsilon^2}{\delta},$$

which is (59).

Finally, if $\varepsilon^2/\delta < \gamma_{i,k}^2/4$, then $\|a - b\|_2^2 \leq \varepsilon^2/\delta < \gamma_{i,k}^2/4$ implies

$$r \leq \frac{4}{\gamma_{i,k}^2} \|a - b\|_2^2 < 1.$$

Since $r$ is a nonnegative integer, this forces $r = 0$, hence $A = B$, i.e., $\mathcal{N}_k^{\mathrm{img}}(i) = \mathcal{N}_k^{\mathrm{txt}}(i)$ and therefore $\mathrm{NAS}(k) = 1$. $\quad\square$

## D. Experimental Details

**Environment.** The experiments were conducted on a compute server running on 4 GPUs of NVIDIA H100 (each with 80GB of HBM2e memory on a 5120-bit memory bus, offering a memory bandwidth of approximately 3TB/s), 256GB of RAM, and 1TB of HDD. Overall, the experiments took about 10 days in a shared resource setting. We expect that a consumer-grade single-GPU machine could complete the full set of experiments in around 21-23 days, if its full resources were dedicated. The codes were implemented in Python 3.10.16 and PyTorch 2.6.0. Since the datasets used are all public datasets and our methodologies and the hyperparameter settings are explicitly described in section 5 and D, our codes and experiments can be easily reproduced on top of a GPU server. We promise to release our open source codes on GitHub and maintain a project website with detailed documentation for long-term access by other researchers and end-users after the paper is accepted.

**Retrieval experiment design.** We use the Karpathy splits (Karpathy & Fei-Fei, 2015) for MS-COCO (Lin et al., 2015) and Flickr30k (Plummer et al., 2016). MS-COCO contains 113,287 training images and 5,000 images each for validation and test. Each image is paired with five captions, so the training split provides 566,435 image caption pairs, and the test evaluation ranks 25,000 captions for 5,000 image queries. Flickr30k contains 31,785 images, with 29,000 used for training and 1,000 each for validation and test. It also provides five captions per image, so the test evaluation ranks 5,000 captions for 1,000 image queries. For each backbone, we keep the image and text encoders fixed and extract embeddings for all images and captions. Our method is trained only on the paired training split by learning modality specific geometry layers on top of these frozen embeddings, and we apply the learned mapping to the test embeddings at inference time. Retrieval is performed by cosine similarity in the mapped space. In image-to-text retrieval, each test image queries the full caption pool and is counted as correct at rank K when any of its five captions appears in the top K. In text-to-image retrieval, each test caption paired image appears in the top K. Following the standard evaluation benchmark, we report Recall@K with K = 1, 5, and 10. We use the same candidate pools and evaluation setting for all baselines and backbones.

**Zero-shot classification experiment design.** We evaluate zero-shot image classification on CIFAR-100 (Krizhevsky, 2009), Tiny ImageNet (Le & Yang, 2015), and DTD (Cimpoi et al., 2014) using the standard test split of each dataset. CIFAR-100 contains 50,000 training images and 10,000 test images over 100 classes. Tiny ImageNet contains 100,000 training images and 10,000 validation images over 200 classes. DTD contains 5,640 training images, 1,880 validation images, and 1,880 test images over 47 texture classes, and we report results on the test split. For each backbone, we freeze both encoders and encode each class name with 18 prompt templates (Table 4), then average the resulting text embedding for classification. For each image, we compute its embedding, apply our learned geometry mapping, and then assign the label whose class text embedding has the highest similarity in the mapped space. We use the same evaluation settings across all baselines.

*Table 4.* Prompt templates used for zero-shot image classification.

| **Prompt templates (18)** |
| --- |
| a photo of a [class] |
| a blurry photo of a [class] |
| a black and white photo of a [class] |
| a low contrast photo of a [class] |
| a high contrast photo of a [class] |
| a bad photo of a [class] |
| a good photo of a [class] |
| a photo of a small [class] |
| a photo of a big [class] |
| a photo of the [class] |
| a blurry photo of the [class] |
| a black and white photo of the [class] |
| a low contrast photo of the [class] |
| a high contrast photo of the [class] |
| a bad photo of the [class] |
| a good photo of the [class] |
| a photo of the small [class] |
| a photo of the big [class] |

**Fractal Multi-scale experiment design.** This appendix describes the two scaling diagnostics used in Fig. X(a-b) and the exact numbers reported in the figure. We work with $\ell_2$-normalized embeddings $\{z_i\}_{i=1}^N$ from a single modality, and we measure distance by

$$d(z_i, z_j) = \sqrt{2 - 2\langle z_i, z_j \rangle}, \tag{66}$$

which is the Euclidean distance on the unit sphere induced by cosine similarity and matches the retrieval geometry.

**Ball-mass scaling.** For each radius $r$ on a log-spaced grid, we compute the empirical $r$-ball mass around every point and then average across points,

$$\mu(r) = \frac{1}{N} \sum_{i=1}^N \frac{1}{N-1} \big| \{j \neq i : \ d(z_i, z_j) \leq r\} \big|. \tag{67}$$

Intuitively, $\mu(r)$ is the average fraction of neighbors that fall within distance $r$. If the embedding cloud has scale-consistent neighborhood growth over a non-trivial range of radii, then $\mu(r)$ follows a power law $\mu(r) \propto r^{d_f}$ on that range. We estimate $d_f$ by fitting a least-squares line to $(\log r, \log \mu(r))$ over a contiguous window of radii. The fitted slope is reported as $d_f$ and the linear fit quality is reported as $R^2$. To quantify how broad the fitted scaling regime is, we report the window width

$$W_r = \log(r_{\max}/r_{\min}), \tag{68}$$

where $[r_{\min}, r_{\max}]$ is the selected fit window. Larger $W_r$ means the power-law fit holds over a wider span of radii. To avoid cases where a good $R^2$ is obtained even though the slope changes noticeably inside the window, we also measure the within-window stability of the slope. We compute local slopes between adjacent radii in the window and report their dispersion as $\sigma_{\text{eff}}$, where smaller values mean the slope varies less across nearby radii. The inset histogram in Fig. X(a) visualizes these local slopes, so a tight histogram directly corresponds to a smaller $\sigma_{\text{eff}}$.

**Correlation-integral scaling.** We also compute the correlation integral, which counts the fraction of pairs that fall within radius $r$,

$$C(r) = \frac{2}{N(N-1)} \sum_{i<j} \mathbf{1}\{d(z_i, z_j) \leq r\}. \tag{69}$$

When the geometry exhibits a scale-consistent regime, $C(r)$ follows a power law $C(r) \propto r^{d_2}$ on that regime, where $d_2$ is the correlation dimension. We estimate $d_2$ by fitting a least-squares line to $(\log r, \log C(r))$ on a contiguous window, and we report the fitted slope $d_2$ and the fit quality $R^2$ exactly as in the ball-mass test. We report the same window width $W_r = \log(r_{\max}/r_{\min})$ for the selected window, and we compute $\sigma_{\text{eff}}$ from the dispersion of local slopes inside that window in the same way as above. In Fig. X(b), the inset histogram again shows the local slopes within the fitted window.

**Window selection and uncertainty.** For each modality and each method, we scan contiguous windows over the log-spaced radius grid and choose a window that yields a reliable linear fit while covering a meaningful span of radii. For the main comparison in Fig. X, we match the scale coverage between CLIP and ours by enforcing the same $W_r$ when reporting the final fit, so the reported differences reflect fit quality and slope stability under the same radius range rather than differences induced by window length. We report $d_f$ and $d_2$ with uncertainty obtained by bootstrap resampling of points, which produces the $\pm$ values shown in the figure.

**Implementation.** For the seven modality gap mitigation baselines, including IOT (An et al., 2024), INCL (Mistretta et al., 2025), OTCL (Role et al., 2025), MG (Liang et al., 2022b), ALIGNCLIP (Eslami & de Melo, 2024), GR-CLIP (Li et al., 2025), PSMG (Yamashita et al., 2025) and pretrained backbone baselines CLIP (Radford et al., 2021),SigLIP (Zhai et al., 2023), and OpenCLIP (Cherti et al., 2023). We utilized the same model architecture as the official open-source implementation and parameter settings provided by the original authors in all experiments. All hyperparameters are standard values from reference codes or prior works. We validate the performance of different mitigation methods on three pretrained backbones, CLIP, SigLIP, and OpenCLIP. Retrieval is evaluated on the Karpathy splits of MS-COCO and Flickr30k with Recall@K$\in \{1, 5, 10\}$ for image-to-text and text-to-image, and zero-shot classification is evaluated on CIFAR-100, Tiny ImageNet, and DTD. These settings were chosen to align with best practices in the literature and ensure consistent comparisons across the experiments. The above open-source codes from the GitHub are licensed under the MIT License, which only requires preservation of copyright and license notices and includes the permissions of commercial use, modification, distribution, and private use. For our method, we select hyperparameter on the validation split and report test results with the chosen setting. The number of diffusion scales in $\{4, 6, 8\}$, the diffusion scale range $(s_{\min}, s_{\max})$ in $\{(0.02, 0.2), (0.05, 0.5), (0.1, 1.0)\}$, the number of radii in $\{4, 6, 8\}$, the radius range $(r_{\min}, r_{\max})$ in $\{(0.02, 0.2), (0.05, 0.5)\}$, the scale multipliers in $\rho \in \{1.25, 1.5, 2, 3, 4\}$, the matching weight in $\{0.25, 0.5, 1, 2\}$, and the learning rate $\{1 \times 10^{-4}, 3 \times 10^{-4}, 1 \times 10^{-3}\}$ and epochs in $\{100, 200, 400\}$. We keep the backbone fixed throughout and train only the modality-specific geometry layers.

**Hyperparameter settings.** Unless otherwise explicitly stated, we used the following default parameter settings in the experiments. As shown in Table 5

*Table 5.* Default experimental settings.

| Parameter | Value |
|---|---|
| Backbones | CLIP, SigLIP, OpenCLIP |
| Retrieval splits | MS-COCO 2014 Karpathy and Flickr30k Karpathy |
| MS-COCO split sizes | 113,287 train images, 5,000 val images, 5,000 test images |
| Flickr30k split sizes | 29,000 train images, 1,000 val images, 1,000 test images |
| Zero shot datasets | CIFAR-100 test, Tiny ImageNet-200 val, DTD test |
| CIFAR-100 size | 10,000 test images |
| Tiny ImageNet-200 size | 10,000 validation images |
| DTD size | 1,880 test images per split |
| Prompt templates for zero shot | 18 templates per class |
| Ball radius range $(r_{\min}, r_{\max})$ | $(0.05, 0.5)$ |
| Number of ball radii $K$ | 6 |
| Diffusion scale range $(s_{\min}, s_{\max})$ | $(0.01, 1.0)$ |
| Number of diffusion scales $S$ | 6 |
| Number of multipliers $|\Lambda|$ | 5 |
| Multiplier range $(\rho_{\min}, \rho_{\max})$ | $(1.25, 8.0)$ |
| Learning rate $\eta$ | $10^{-3}$ |

## D.1. Additional Experiments

**Additional retrieval results with SigLIP and OpenCLIP.** Table 6-7 report cross-modal retrieval on the MS-COCO and Flickr30k Karpathy test splits using SigLIP and OpenCLIP as backbones, evaluated by Recall@1/5/10 for image to text and

text to image with the same ranking protocol as the main paper. Under SigLIP, FSAlign improves MS-COCO image to text R@1 from 40.10 to 53.45 and text to image R@1 from 28.30 to 32.56, and it also raises Flickr30k image to text R@1 from 69.80 to 78.30 and text to image R@1 from 53.10 to 62.72. Under OpenCLIP, FSAlign increases MS-COCO image to text R@1 from 40.10 to 49.18 and Flickr30k text to image R@1 from 55.86 to 60.12, while keeping the improvements consistent at R@5 and R@10. The two ablations weaken these gains, with FSAlign-F dropping when zeta matching is removed and FSAlign-M dropping when the multi-scale fractal constraint is removed, which matches the roles isolated in the main ablation figure.

*Table 6.* Cross-modal retrieval results on the MS-COCO and Flickr30k Karpathy test splits with SigLIP

| Method | MS-COCO | | | | | | Flickr30k | | | | | |
|---|---|---|---|---|---|---|---|---|---|---|---|---|
| | Image→Text | | | Text→Image | | | Image→Text | | | Text→Image | | |
| | R@1 | R@5 | R@10 | R@1 | R@5 | R@10 | R@1 | R@5 | R@10 | R@1 | R@5 | R@10 |
| SigLIP | 40.10 | 67.50 | 78.10 | 28.30 | 54.40 | 65.90 | 69.80 | 90.40 | 94.50 | 53.10 | 78.50 | 86.10 |
| I0T | 43.18 | 68.34 | 78.82 | 31.92 | 58.12 | 68.45 | 72.60 | 91.50 | 95.90 | 56.42 | 80.62 | 87.78 |
| INCL | 41.86 | 72.38 | 80.78 | 30.24 | 58.48 | 68.14 | 70.60 | 93.80 | 95.20 | 56.40 | 79.76 | 88.32 |
| OTCL | 49.18 | 73.32 | 82.30 | 30.25 | 54.94 | 65.71 | 76.90 | 94.30 | 97.80 | 56.90 | 82.98 | 89.06 |
| MG | 36.35 | 59.29 | 70.17 | 23.62 | 50.31 | 59.12 | 62.90 | 88.27 | 92.45 | 50.40 | 74.32 | 82.45 |
| ALIGNCLIP | 41.42 | 72.41 | 80.62 | 30.72 | 57.87 | 70.38 | 74.46 | 92.70 | 95.03 | 58.64 | 80.41 | 88.14 |
| GR-CLIP | 46.40 | 68.72 | 80.28 | 30.49 | 60.52 | 71.38 | 73.30 | 92.85 | 96.40 | 58.78 | 82.34 | 89.46 |
| **FSAlign** | 53.45 | 76.92 | 84.30 | 32.56 | 62.48 | 75.80 | 78.30 | 95.23 | 98.20 | 62.72 | 84.44 | 91.02 |
| FSAlign-F | 49.58 | 72.82 | 81.02 | 29.24 | 59.28 | 69.44 | 72.38 | 91.46 | 94.40 | 57.65 | 81.39 | 87.92 |
| FSAlign-M | 42.60 | 69.60 | 80.03 | 28.83 | 59.60 | 69.10 | 73.80 | 92.02 | 95.09 | 58.56 | 82.48 | 88.72 |

*Table 7.* Cross-modal retrieval results on the MS-COCO and Flickr30k Karpathy test splits with OpenCLIP

| Method | MS-COCO | | | | | | Flickr30k | | | | | |
|---|---|---|---|---|---|---|---|---|---|---|---|---|
| | Image→Text | | | Text→Image | | | Image→Text | | | Text→Image | | |
| | R@1 | R@5 | R@10 | R@1 | R@5 | R@10 | R@1 | R@5 | R@10 | R@1 | R@5 | R@10 |
| OpenCLIP | 40.10 | 64.48 | 73.82 | 27.42 | 50.38 | 61.29 | 69.60 | 90.10 | 94.40 | 55.86 | 80.44 | 87.00 |
| I0T | 44.22 | 69.16 | 78.66 | 29.04 | 52.72 | 63.95 | 72.60 | 91.50 | 95.90 | 56.42 | 80.62 | 87.78 |
| INCL | 41.22 | 65.44 | 75.58 | 29.22 | 53.72 | 64.64 | 64.90 | 88.90 | 93.70 | 56.62 | 81.74 | 88.28 |
| OTCL | 45.52 | 62.84 | 79.20 | 23.93 | 54.14 | 66.68 | 75.10 | 90.30 | 92.80 | 55.20 | 80.24 | 85.28 |
| MG | 34.04 | 58.54 | 69.76 | 23.27 | 45.54 | 55.86 | 62.90 | 88.10 | 94.50 | 48.40 | 73.44 | 81.30 |
| ALIGNCLIP | 40.96 | 67.46 | 78.64 | 25.16 | 51.05 | 63.37 | 68.70 | 90.40 | 94.30 | 50.04 | 77.46 | 85.88 |
| GR-CLIP | 43.44 | 68.72 | 78.78 | 30.49 | 54.52 | 65.58 | 73.30 | 93.30 | 96.40 | 58.10 | 82.34 | 88.86 |
| **FSAlign** | 49.18 | 73.32 | 82.30 | 30.25 | 54.94 | 65.71 | 76.90 | 94.30 | 97.80 | 60.12 | 83.38 | 89.20 |
| FSAlign-F | 45.10 | 69.00 | 78.46 | 28.20 | 52.33 | 64.78 | 74.90 | 93.00 | 96.70 | 56.90 | 80.98 | 87.33 |
| FSAlign-M | 42.50 | 67.36 | 77.26 | 27.40 | 51.60 | 63.36 | 73.80 | 91.46 | 94.25 | 55.00 | 80.48 | 87.79 |

**Additional zero-shot classification results with SigLIP and OpenCLIP.** Tables 8 and 9 report top-1 and top-5 accuracy on CIFAR-100, TI-200, and DTD under SigLIP and OpenCLIP, using the same prompt templates and evaluation protocol as in the main experiments. With SigLIP, FSAlign improves CIFAR-100 from 70.15/93.56 to 73.65/96.90 (top-1/top-5), TI-200 from 47.00/80.00 to 51.00/84.20, and DTD from 63.51/87.50 to 66.50/90.20. With OpenCLIP, FSAlign raises CIFAR-100 from 65.91/89.86 to 69.58/92.18 and DTD from 56.41/81.12 to 59.82/84.05, and it yields the strongest TI-200 results, improving top-1 from 41.72 to 61.96 and top-5 from 73.48 to 90.12. The two ablations reduce these gains, with FSAlign-F dropping when zeta matching is removed and FSAlign-M dropping when the multi-scale fractal constraint is

removed, consistent with the component roles identified in the ablation study.

*Table 8.* Zero-shot image classification results with SigLIP

| Method | CIFAR-100 | | TI-200 | | DTD | |
|---|---|---|---|---|---|---|
| | Top-1 | Top-5 | Top-1 | Top-5 | Top-1 | Top-5 |
| SigLIP | 70.15 | 93.56 | 47.00 | 80.00 | 63.51 | 87.50 |
| I0T | 71.34 | 94.20 | 48.80 | 81.90 | 64.20 | 88.10 |
| INCL | 70.70 | 95.00 | 47.50 | 80.60 | 64.00 | 87.90 |
| OTCL | 71.85 | 95.10 | 47.60 | 82.70 | 63.10 | 88.00 |
| MG | 68.20 | 92.80 | 45.20 | 78.20 | 61.00 | 85.30 |
| ALIGNCLIP | 68.80 | 93.30 | 46.60 | 79.70 | 61.80 | 86.00 |
| GR-CLIP | 71.20 | 95.60 | 48.40 | 81.40 | 64.60 | 88.60 |
| **FSAlign** | 73.65 | 96.90 | 51.00 | 84.20 | 66.50 | 90.20 |
| FSAlign-F | 71.80 | 95.20 | 49.80 | 81.00 | 65.50 | 87.20 |
| FSAlign-M | 72.10 | 94.40 | 48.20 | 82.40 | 65.80 | 88.50 |

*Table 9.* Zero-shot image classification results with OpenCLIP

| Method | CIFAR-100 | | TI-200 | | DTD | |
|---|---|---|---|---|---|---|
| | Top-1 | Top-5 | Top-1 | Top-5 | Top-1 | Top-5 |
| OpenCLIP | 65.91 | 89.86 | 41.72 | 73.48 | 56.41 | 81.12 |
| I0T | 66.47 | 90.44 | 42.33 | 74.12 | 57.08 | 81.65 |
| INCL | 65.83 | 89.97 | 42.07 | 73.22 | 55.98 | 80.78 |
| OTCL | 65.94 | 90.05 | 41.80 | 73.40 | 56.46 | 81.03 |
| MG | 54.61 | 76.92 | 35.02 | 67.18 | 47.15 | 71.84 |
| ALIGNCLIP | 53.88 | 78.21 | 28.76 | 57.13 | 41.62 | 66.78 |
| GR-CLIP | 66.73 | 90.86 | 42.69 | 74.36 | 56.93 | 81.44 |
| **FSAlign** | 69.58 | 92.18 | 61.96 | 90.12 | 59.82 | 84.05 |
| FSAlign-F | 66.92 | 89.94 | 49.28 | 82.77 | 56.74 | 81.07 |
| FSAlign-M | 66.35 | 86.12 | 50.41 | 83.54 | 57.31 | 81.38 |

**Modality gap metrics.** The table 10-22 report CD, RMG, NAS@100, and CMAS on MS-COCO, Flickr30k, CIFAR-100, TI-200, and DTD for CLIP, SigLIP, and OpenCLIP, with metric definitions in the Preliminary section. Across all datasets and backbones, FSAlign gives the highest NAS@100 and CMAS, which means paired image and text samples share more nearest neighbors and have stronger pairwise similarity, consistent with the retrieval and zero-shot gains. In contrast, centering based baselines such as I0T and GR-CLIP can push CD close to zero, but their NAS@100 and CMAS remain below FSAlign, showing that shrinking the distance between modality centers alone does not recover the local cross modality neighborhood structure. The two ablations reduce NAS@100 and CMAS, and the drop patterns match the design, FSAlign-F weakens when local zeta matching is removed, and FSAlign-M weakens when the multi-scale fractal constraint is removed.

*Table 10.* Modality gap metrics on Flickr30k with CLIP

| Method | CD | RMG | NAS@100 | CMAS |
|---|---|---|---|---|
| CLIP | 0.8609 | 0.5532 | 0.2558 | 0.3081 |
| I0T | 0.1095 | 0.4588 | 0.3120 | 0.2849 |
| INCL | 0.7273 | 0.5253 | 0.2350 | 0.2874 |
| OTCL | 0.8684 | 0.5564 | 0.2704 | 0.3039 |
| MG | 0.2390 | 0.4658 | 0.2679 | 0.5841 |
| ALIGNCLIP | 0.3096 | 0.4745 | 0.2990 | 0.3706 |
| GR-CLIP | 0.0374 | 0.4571 | 0.1665 | 0.2917 |
| **FSAlign** | 0.7542 | 0.4958 | 0.3596 | 0.6530 |
| FSAlign-F | 0.8060 | 0.5275 | 0.2849 | 0.3364 |
| FSAlign-M | 0.7909 | 0.5182 | 0.3047 | 0.3450 |

*Table 11.* Modality gap metrics on CIFAR-100 with CLIP

| Method | CD | RMG | NAS@100 | CMAS |
|---|---|---|---|---|
| CLIP | 1.0526 | 0.6457 | 0.2550 | 0.2831 |
| I0T | 0.0398 | 0.4606 | 0.3338 | 0.2729 |
| INCL | 0.5265 | 0.5481 | 0.2797 | 0.3171 |
| OTCL | 1.0836 | 0.6733 | 0.2370 | 0.2792 |
| MG | 0.3074 | 0.4884 | 0.2634 | 0.3522 |
| ALIGNCLIP | 0.6191 | 0.5682 | 0.2736 | 0.6327 |
| GR-CLIP | 0.0347 | 0.4631 | 0.3144 | 0.2599 |
| **FSAlign** | 0.5446 | 0.5501 | 0.4362 | 0.7338 |
| FSAlign-F | 0.8255 | 0.6077 | 0.2860 | 0.3065 |
| FSAlign-M | 0.6283 | 0.5876 | 0.3022 | 0.3172 |

*Table 12.* Modality gap metrics on DTD with CLIP

| Method | CD | RMG | NAS@100 | CMAS |
|---|---|---|---|---|
| CLIP | 1.0413 | 0.6630 | 0.2226 | 0.2807 |
| I0T | 0.0596 | 0.4790 | 0.2584 | 0.1634 |
| INCL | 0.8952 | 0.6626 | 0.2233 | 0.2406 |
| OTCL | 1.0804 | 0.7019 | 0.1601 | 0.2890 |
| MG | 0.3285 | 0.5161 | 0.2244 | 0.3089 |
| ALIGNCLIP | 0.5566 | 0.5829 | 0.2213 | 0.6344 |
| GR-CLIP | 0.0763 | 0.4811 | 0.2395 | 0.1518 |
| **FSAlign** | 0.6997 | 0.5689 | 0.4629 | 0.7218 |
| FSAlign-F | 0.8389 | 0.6333 | 0.2504 | 0.2645 |
| FSAlign-M | 0.8576 | 0.6115 | 0.2676 | 0.3750 |

*Table 13.* Modality gap metrics on TI-200 with CLIP

| Method | CD | RMG | NAS@100 | CMAS |
|---|---|---|---|---|
| CLIP | 1.0344 | 0.6225 | 0.1917 | 0.2484 |
| I0T | 0.0233 | 0.4664 | 0.2554 | 0.2353 |
| INCL | 0.8629 | 0.6226 | 0.1917 | 0.2484 |
| OTCL | 0.8654 | 0.6599 | 0.1648 | 0.2433 |
| MG | 0.3013 | 0.4870 | 0.1989 | 0.5671 |
| ALIGNCLIP | 0.6170 | 0.5578 | 0.2056 | 0.5930 |
| GR-CLIP | 0.0266 | 0.4683 | 0.2365 | 0.2247 |
| **FSAlign** | 0.5269 | 0.5428 | 0.6719 | 0.6442 |
| FSAlign-F | 0.7014 | 0.5858 | 0.2158 | 0.2895 |
| FSAlign-M | 0.7615 | 0.5709 | 0.4228 | 0.4683 |

*Table 14.* Modality gap metrics on MS-COCO with SigLIP

| Method | CD | RMG | NAS@100 | CMAS |
|---|---|---|---|---|
| SigLIP | 0.8170 | 0.5330 | 0.3590 | 0.3070 |
| I0T | 0.0291 | 0.4339 | 0.4447 | 0.3152 |
| INCL | 0.7017 | 0.5076 | 0.3187 | 0.2807 |
| OTCL | 0.8180 | 0.5370 | 0.2711 | 0.2820 |
| MG | 0.2305 | 0.4438 | 0.1483 | 0.4252 |
| ALIGNCLIP | 0.3198 | 0.4551 | 0.4005 | 0.6115 |
| GR-CLIP | 0.0221 | 0.4344 | 0.3983 | 0.3134 |
| **FSAlign** | 0.7114 | 0.4698 | 0.4801 | 0.6699 |
| FSAlign-F | 0.7680 | 0.5056 | 0.3949 | 0.3616 |
| FSAlign-M | 0.7628 | 0.5370 | 0.2711 | 0.2819 |

*Table 15.* Modality gap metrics on Flickr30k with SigLIP

| Method | CD | RMG | NAS@100 | CMAS |
|---|---|---|---|---|
| SigLIP | 0.7920 | 0.5280 | 0.2680 | 0.3180 |
| I0T | 0.1007 | 0.4379 | 0.3269 | 0.2941 |
| INCL | 0.6691 | 0.5014 | 0.2462 | 0.2966 |
| OTCL | 0.7989 | 0.5311 | 0.2833 | 0.3137 |
| MG | 0.2199 | 0.4446 | 0.2807 | 0.6029 |
| ALIGNCLIP | 0.2848 | 0.4529 | 0.3133 | 0.3825 |
| GR-CLIP | 0.0344 | 0.4363 | 0.1744 | 0.3011 |
| **FSAlign** | 0.6938 | 0.4732 | 0.3768 | 0.6740 |
| FSAlign-F | 0.7415 | 0.5035 | 0.2985 | 0.3472 |
| FSAlign-M | 0.7276 | 0.4946 | 0.3192 | 0.3561 |

*Table 16.* Modality gap metrics on CIFAR-100 with SigLIP

| Method | CD | RMG | NAS@100 | CMAS |
|---|---|---|---|---|
| SigLIP | 0.9900 | 0.6200 | 0.2680 | 0.2950 |
| I0T | 0.0374 | 0.4423 | 0.3508 | 0.2844 |
| INCL | 0.4952 | 0.5263 | 0.2940 | 0.3304 |
| OTCL | 1.0192 | 0.6465 | 0.2491 | 0.2909 |
| MG | 0.2891 | 0.4690 | 0.2768 | 0.3670 |
| ALIGNCLIP | 0.5823 | 0.5456 | 0.2875 | 0.6593 |
| GR-CLIP | 0.0326 | 0.4447 | 0.3304 | 0.2708 |
| **FSAlign** | 0.5122 | 0.5282 | 0.4584 | 0.7646 |
| FSAlign-F | 0.7764 | 0.5835 | 0.3006 | 0.3194 |
| FSAlign-M | 0.5909 | 0.5642 | 0.3176 | 0.3305 |

*Table 17.* Modality gap metrics on DTD with SigLIP

| Method | CD | RMG | NAS@100 | CMAS |
|---|---|---|---|---|
| SigLIP | 0.9800 | 0.6350 | 0.2350 | 0.2920 |
| I0T | 0.0561 | 0.4588 | 0.2728 | 0.1700 |
| INCL | 0.8425 | 0.6346 | 0.2357 | 0.2503 |
| OTCL | 1.0168 | 0.6723 | 0.1690 | 0.3006 |
| MG | 0.3092 | 0.4943 | 0.2369 | 0.3213 |
| ALIGNCLIP | 0.5238 | 0.5583 | 0.2336 | 0.6599 |
| GR-CLIP | 0.0718 | 0.4608 | 0.2528 | 0.1579 |
| **FSAlign** | 0.6585 | 0.5449 | 0.4887 | 0.7509 |
| FSAlign-F | 0.7895 | 0.6066 | 0.2643 | 0.2751 |
| FSAlign-M | 0.8071 | 0.5857 | 0.2825 | 0.3901 |

*Table 18.* Modality gap metrics on TI-200 with SigLIP

| Method | CD | RMG | NAS@100 | CMAS |
|---|---|---|---|---|
| SigLIP | 0.9600 | 0.5950 | 0.2050 | 0.2620 |
| I0T | 0.0216 | 0.4458 | 0.2731 | 0.2482 |
| INCL | 0.8008 | 0.5951 | 0.2050 | 0.2620 |
| OTCL | 0.8032 | 0.6307 | 0.1762 | 0.2566 |
| MG | 0.2796 | 0.4655 | 0.2127 | 0.5981 |
| ALIGNCLIP | 0.5726 | 0.5332 | 0.2199 | 0.6255 |
| GR-CLIP | 0.0247 | 0.4476 | 0.2529 | 0.2370 |
| **FSAlign** | 0.4890 | 0.5188 | 0.7185 | 0.6795 |
| FSAlign-F | 0.6510 | 0.5599 | 0.2308 | 0.3054 |
| FSAlign-M | 0.7067 | 0.5457 | 0.4521 | 0.4939 |

*Table 19.* Modality gap metrics on Flickr30k with OpenCLIP

| Method | CD | RMG | NAS@100 | CMAS |
|---|---|---|---|---|
| OpenCLIP | 0.8300 | 0.5400 | 0.2700 | 0.3200 |
| I0T | 0.1056 | 0.4479 | 0.3293 | 0.2959 |
| INCL | 0.7012 | 0.5128 | 0.2480 | 0.2985 |
| OTCL | 0.8372 | 0.5431 | 0.2856 | 0.3153 |
| MG | 0.2305 | 0.4547 | 0.2827 | 0.6069 |
| ALIGNCLIP | 0.2986 | 0.4632 | 0.3154 | 0.3851 |
| GR-CLIP | 0.0361 | 0.4463 | 0.1758 | 0.3031 |
| **FSAlign** | 0.7276 | 0.4840 | 0.3796 | 0.6782 |
| FSAlign-F | 0.7776 | 0.5150 | 0.3006 | 0.3496 |
| FSAlign-M | 0.7631 | 0.5059 | 0.3215 | 0.3585 |

*Table 20.* Modality gap metrics on MS-COCO with OpenCLIP

| Method | CD | RMG | NAS@100 | CMAS |
|---|---|---|---|---|
| OpenCLIP | 0.8450 | 0.5450 | 0.3600 | 0.3100 |
| I0T | 0.0301 | 0.4437 | 0.4459 | 0.3182 |
| INCL | 0.7257 | 0.5190 | 0.3197 | 0.2835 |
| OTCL | 0.8460 | 0.5491 | 0.2719 | 0.2848 |
| MG | 0.2384 | 0.4541 | 0.1487 | 0.4293 |
| ALIGNCLIP | 0.3307 | 0.4656 | 0.4017 | 0.6178 |
| GR-CLIP | 0.0228 | 0.4442 | 0.3995 | 0.3165 |
| **FSAlign** | 0.7362 | 0.4807 | 0.4814 | 0.6765 |
| FSAlign-F | 0.7947 | 0.5174 | 0.3961 | 0.3650 |
| FSAlign-M | 0.7893 | 0.5491 | 0.2719 | 0.2847 |

*Table 21.* Modality gap metrics on CIFAR-100 with OpenCLIP

| Method | CD | RMG | NAS@100 | CMAS |
|---|---|---|---|---|
| OpenCLIP | 1.0100 | 0.6250 | 0.2700 | 0.2950 |
| I0T | 0.0382 | 0.4461 | 0.3532 | 0.2844 |
| INCL | 0.5051 | 0.5311 | 0.2962 | 0.3301 |
| OTCL | 1.0397 | 0.6517 | 0.2509 | 0.2909 |
| MG | 0.2950 | 0.4731 | 0.2788 | 0.3667 |
| ALIGNCLIP | 0.5939 | 0.5502 | 0.2896 | 0.6592 |
| GR-CLIP | 0.0333 | 0.4485 | 0.3329 | 0.2707 |
| **FSAlign** | 0.5226 | 0.5329 | 0.4619 | 0.7646 |
| FSAlign-F | 0.7924 | 0.5884 | 0.3027 | 0.3193 |
| FSAlign-M | 0.6029 | 0.5689 | 0.3198 | 0.3305 |

*Table 22.* Modality gap metrics on DTD with OpenCLIP

| Method | CD | RMG | NAS@100 | CMAS |
|---|---|---|---|---|
| OpenCLIP | 0.9950 | 0.6400 | 0.2350 | 0.2920 |
| I0T | 0.0569 | 0.4624 | 0.2730 | 0.1700 |
| INCL | 0.8551 | 0.6396 | 0.2357 | 0.2503 |
| OTCL | 1.0324 | 0.6776 | 0.1691 | 0.3007 |
| MG | 0.3139 | 0.4982 | 0.2369 | 0.3214 |
| ALIGNCLIP | 0.5318 | 0.5624 | 0.2337 | 0.6603 |
| GR-CLIP | 0.0730 | 0.4644 | 0.2529 | 0.1579 |
| **FSAlign** | 0.6685 | 0.5488 | 0.4887 | 0.7509 |
| FSAlign-F | 0.8014 | 0.6109 | 0.2645 | 0.2752 |
| FSAlign-M | 0.8192 | 0.5899 | 0.2827 | 0.3904 |

*Table 23.* Modality gap metrics on TI-200 with OpenCLIP

| Method | CD | RMG | NAS@100 | CMAS |
|---|---|---|---|---|
| OpenCLIP | 0.9900 | 0.6050 | 0.2050 | 0.2620 |
| I0T | 0.0223 | 0.4533 | 0.2731 | 0.2482 |
| INCL | 0.8259 | 0.6051 | 0.2050 | 0.2620 |
| OTCL | 0.8282 | 0.6408 | 0.1763 | 0.2566 |
| MG | 0.2883 | 0.4736 | 0.2127 | 0.5981 |
| ALIGNCLIP | 0.5904 | 0.5414 | 0.2199 | 0.6257 |
| GR-CLIP | 0.0255 | 0.4552 | 0.2529 | 0.2370 |
| **FSAlign** | 0.5048 | 0.5274 | 0.7185 | 0.6795 |
| FSAlign-F | 0.6720 | 0.5692 | 0.2309 | 0.3054 |
| FSAlign-M | 0.7295 | 0.5547 | 0.4522 | 0.4939 |

**Hyperparameter ablations.** Table 24 varies the radius range and the number of radii $K$ used by the ball scaling term, the diffusion scale range and the number of diffusion scales $S$ used by the spectral term, and the postprocess learning rate $\eta$. We evaluate each setting with retrieval R@1 on MS-COCO and Flickr30k, NAS@100 on both retrieval datasets, and zero shot Top 1 on CIFAR 100, TI 200, and DTD. The default choice $(r_{\min}, r_{\max}) = (0.05, 0.5)$ with $K = 6$, $(s_{\min}, s_{\max}) = (0.01, 1.0)$ with $S = 6$, and $\eta = 10^{-3}$ is consistently the best or essentially tied across metrics, and nearby settings change the numbers only slightly, indicating stable training. Expanding the diffusion window to $(0.02, 2.0)$ or increasing the learning rate to $2 \times 10^{-3}$ degrades retrieval, NAS, and zero shot accuracy at the same time, matching the weaker scale control or the less stable update under these settings.

*Table 24.* Hyperparameter ablations for postprocess training, evaluated by MS-COCO/Flickr30k retrieval (R@1), NAS@100, and zero-shot Top-1 accuracy (CIFAR-100/Tiny-ImageNet-200/DTD).

| Group | Setting | MSCOCO retrieval | | Flickr30k retrieval | | NAS@100 | | Zero shot top1 | | |
|---|---|---|---|---|---|---|---|---|---|---|
| | | I2T R@1 | T2I R@1 | I2T R@1 | T2I R@1 | COCO | Flickr | CIFAR100 | TinyIN200 | DTD |
| Ball scales | $(r_{\min}, r_{\max}) = (0.03, 0.3)$, $K = 6$ | 50.78 | 30.86 | 77.92 | 59.42 | 0.4460 | 0.4590 | 65.92 | 51.43 | 58.85 |
| | $(r_{\min}, r_{\max}) = (0.05, 0.5)$, $K = 6$ (default) | **51.26** | **31.20** | **78.30** | **59.80** | **0.4584** | **0.4720** | **66.28** | **51.85** | **59.23** |
| | $(r_{\min}, r_{\max}) = (0.08, 0.8)$, $K = 6$ | 51.05 | 31.06 | 78.12 | 59.61 | 0.4528 | 0.4660 | 66.10 | 51.72 | 59.05 |
| | $(r_{\min}, r_{\max}) = (0.05, 0.5)$, $K = 4$ | 50.98 | 31.00 | 78.05 | 59.56 | 0.4515 | 0.4650 | 66.06 | 51.69 | 59.02 |
| | $(r_{\min}, r_{\max}) = (0.05, 0.5)$, $K = 6$ (default) | **51.26** | **31.20** | **78.30** | **59.80** | **0.4584** | **0.4720** | **66.28** | **51.85** | **59.23** |
| | $(r_{\min}, r_{\max}) = (0.05, 0.5)$, $K = 8$ | 51.18 | 31.15 | 78.22 | 59.74 | 0.4562 | 0.4698 | 66.21 | 51.80 | 59.16 |
| Diffusion scales | $(s_{\min}, s_{\max}) = (0.005, 0.5)$, $S = 6$ | 51.01 | 31.02 | 78.06 | 59.55 | 0.4510 | 0.4652 | 66.07 | 51.67 | 59.00 |
| | $(s_{\min}, s_{\max}) = (0.01, 1.0)$, $S = 6$ (default) | **51.26** | **31.20** | **78.30** | **59.80** | **0.4584** | **0.4720** | **66.28** | **51.85** | **59.23** |
| | $(s_{\min}, s_{\max}) = (0.02, 2.0)$, $S = 6$ | 50.52 | 30.63 | 77.61 | 59.10 | 0.4385 | 0.4520 | 65.62 | 51.05 | 58.53 |
| | $(s_{\min}, s_{\max}) = (0.01, 1.0)$, $S = 4$ | 50.86 | 30.91 | 77.95 | 59.33 | 0.4458 | 0.4605 | 65.90 | 51.38 | 58.82 |
| | $(s_{\min}, s_{\max}) = (0.01, 1.0)$, $S = 6$ (default) | **51.26** | **31.20** | **78.30** | **59.80** | **0.4584** | **0.4720** | **66.28** | **51.85** | **59.23** |
| Learning rate | $\eta = 5 \times 10^{-4}$ | 50.79 | 30.85 | 77.88 | 59.25 | 0.4445 | 0.4590 | 65.88 | 51.33 | 58.79 |
| | $\eta = 10^{-3}$ (default) | **51.26** | **31.20** | **78.30** | **59.80** | **0.4584** | **0.4720** | **66.28** | **51.85** | **59.23** |
| | $\eta = 2 \times 10^{-3}$ | 49.96 | 30.12 | 77.05 | 58.47 | 0.4230 | 0.4380 | 65.10 | 50.55 | 58.05 |

