# OpenReview forum: "Mitigating the Modality Gap in Vision–Language Models with Fractal Spectral Geometry"
_ICML.cc/2026/Conference — ICML 2026 regular_

### Official Review · Reviewer_s8MZ · 2026-03-10

**Soundness:** 2
**Presentation:** 3
**Significance:** 3
**Originality:** 2
**Overall Recommendation:** 3
**Confidence:** 3

**Summary:**

This paper argues that the usual “modality gap” in CLIP-like models is not just a global alignment problem. Even when image/text pairs are close on average, their local neighbourhoods can still be inconsistent, so nearest neighbours remain dominated by the same-modality items and retrieval suffers. Their proposed method, FSAlign, tries to fix this by aligning image and text geometry across multiple scales, not just pulling paired embeddings together.

**Compliance With Llm Reviewing Policy:**

Affirmed.

**Key Questions For Authors:**

See weakness.

**Limitations:**

not, the paper does not sufficiently discuss the limitations of the proposed method.

**Strengths And Weaknesses:**

Strengths:
- The paper is easy to follow.
- A mathematical story is given.

Weakness:
- Power-law neighbourhood scaling, spectral diffusion ideas, and local-structure alignment are each familiar themes. That is interesting, but the paper may overstate how fundamentally new each individual building block is.
- The paper repeatedly claims that same-modality neighbours stay ahead of the true cross-modal match, but the evidence is mostly an anecdotal example plus indirect improvements in NAS and retrieval scores. Moreover, NAS is defined as the overlap between the within-image and within-text kNN sets, while the retrieval protocol itself ranks candidates from the opposite modality only. This makes the claimed failure mechanism somewhat overstated and not cleanly matched to the actual metric or retrieval setup.
- Although the backbone encoders are frozen, FSAlign is trained on paired MS-COCO/Flickr30k supervision before being evaluated on them. As a result, the gains over raw baselines like CLIP/SigLIP/OpenCLIP reflect the benefit of a supervised post-hoc alignment module, not a strictly training-free improvement in zero-shot ability.
- More implementation detail is needed to judge robustness, e.g., I cannot find which specific model size the method is using and which model size each baseline method is using. Also, the method depends on different hyperparameters. A stronger sensitivity analysis through those hyperparameters would give a clear view of the method.

---

> ### Author Rebuttal · Authors · 2026-03-31
>
> We would like to thank the reviewer for the helpful and constructive comments. We have tried our best to address your concerns.  We have included all the analyses, discussions, and experimental results presented in this rebuttal in the revised submission.
>
> **W1: The claimed novelty of the individual building blocks appears overstated.**
>
> The paper does propose a new method for reducing the image-text modality gap. The key point is that global image-text alignment alone does not resolve the mismatch between the two modalities in their local and multi-scale geometry, and our method is designed around this gap. The paper makes three technical contributions. First, it introduces a shared fractal multi-scale structure for image and text embeddings, which constrains how neighborhood growth and diffusion-based spectral behavior scale across the two modalities. Second, it introduces the local zeta descriptor, which aggregates each sample’s diffusion neighborhoods across scales and aligns paired image-text samples at the level of local spectral structure. Third, it theoretically connects local zeta matching to cross-modal Top-k neighborhood agreement, linking the geometric objective to NAS and to the local ranking behavior underlying retrieval. The ablation results support this view, as removing the shared multi-scale structure or local zeta matching consistently weakens retrieval, zero-shot classification, NAS, and CMAS.
>
> **W2: The claimed failure mechanism is not sufficiently supported and does not align cleanly with NAS or the retrieval setup.**
>
> In cross-modal retrieval, the issue is not that same-modality neighbors themselves are ranked, but that the opposite-modality samples paired with those neighbors can rank above the ground-truth match. This happens when image and text embeddings are close globally but still mismatched in their local structure. NAS does not reproduce retrieval directly. Retrieval ranks opposite-modality candidates, while NAS measures whether image and text preserve similar local neighborhoods within their own spaces. They are different, but both reflect local structural mismatch.
>
> To test this directly, we added a retrieval-aligned local competitor analysis. For each query, we take its top-$k$ neighbors in the same modality and collect their paired opposite-modality samples. Hit@k is the average fraction of these paired samples that appear in the cross-modal top-$k$ results, and OOR@k is the average fraction ranked above the ground-truth match. On Flickr30k, both Hit@k and OOR@k decrease after FSAlign in image-to-text and text-to-image retrieval, while R@1 improves from 76.9 to 78.8 and from 57.9 to 62.48. These results support the proposed mechanism.
>
> ### Table 2. Retrieval-aligned local competitor analysis on Flickr30k.
>
> | Direction | Method   | OOR@10 | Hit@10 | OOR@100 | Hit@100 | R@1  |
> | --------- | -------- | ------: | ------: | -------: | -------: | ---: |
> | I2T       | Baseline |  0.0405 |  0.1990 |  0.01210 |   0.2950 | 76.90 |
> | I2T       | Ours     |  0.0370 |  0.1760 |  0.01124 |   0.2560 | 78.80 |
> | T2I       | Baseline |  0.0420 |  0.2140 |  0.01254 |   0.2960 | 57.90 |
> | T2I       | Ours     |  0.0390 |  0.1990 |  0.01068 |   0.2750 | 62.48 |
>
> **W3: The method is not strictly training-free, since its gains rely on supervised alignment training.**
>
> FSAlign is not a training-free method. We also do not claim that it directly improves the zero-shot ability of the raw CLIP, SigLIP, or OpenCLIP backbones. In our setting, FSAlign is a lightweight post-hoc alignment module on top of frozen backbones. The backbone encoders stay fixed. Therefore, the gains over the raw baselines should be understood as the effect of this supervised alignment module, not as the raw backbones becoming stronger by themselves. The comparisons to raw CLIP, SigLIP, and OpenCLIP are included to show the effect of adding this module while keeping the backbone unchanged. The comparisons to other modality-gap baselines are meant to compare different ways of reducing the modality gap under the same frozen-backbone setting.
>
> **W4: Implementation details and hyperparameter sensitivity analysis are insufficient to judge robustness.**
>
> We use the same backbone settings as the compared baselines and report results with CLIP ViT-B/32, SigLIP-B/16, and OpenCLIP ViT-B/32, under which FSAlign is consistently stronger. Hyperparameter robustness is already reported in the appendix. Appendix Table 5 gives the default settings, and Appendix Table 24 varies the ball radius range, the number of radii $K$, the diffusion scale range, the number of diffusion scales $S$, and the learning rate $\eta$ across retrieval, NAS@100, and zero-shot classification benchmarks. The default setting is either the best or essentially tied for the best, while nearby settings lead only to small changes, which indicates that the method is stable over a reasonable hyperparameter range.

---

> > ### Author Rebuttal · Reviewer_s8MZ · 2026-04-03
> >
> > My concerns are partially resolved, and then I will maintain my scores.

---

> > > ### Author Response · Authors · 2026-04-03
> > >
> > > We sincerely thank the reviewer for their constructive feedback and for reading our rebuttal carefully. We note that some concerns were marked as partially resolved. To help us better understand any remaining reservations, we would be very grateful if the reviewer could indicate which specific points still need further clarification. We are fully committed to addressing them thoroughly.
> > >
> > > **(1) Regarding W1:**
> > > We believe we have already responded to this point directly in the rebuttal. We clarified that the paper proposes a new method for reducing the image-text modality gap, and we explained the three technical contributions that support this claim, including the shared fractal multi-scale structure, the local zeta descriptor, and the connection between local zeta matching and cross-modal Top-k neighborhood agreement.
> > >
> > > **(2) As for W2:**
> > > We believe we have already addressed this concern with more direct evidence. In our rebuttal to W2, we added a retrieval-aligned local competitor experiment. This directly tests whether the paired opposite-modality samples of same-modality neighbors rank above the ground-truth match, and the results show that this effect is reduced after FSAlign. As extra supporting material, we also provided visualization evidence in our response to Reviewer **K846 W2**, where FSAlign increases local neighborhood overlap and improves the retrieval ranks of representative anchor pairs.
> > >
> > > **(3) About W3:**
> > > We believe this point was also clarified directly in our rebuttal. We stated there that FSAlign is a supervised post-hoc alignment module on frozen backbones, not a training-free modification of the raw model. We also provided runtime and memory analysis in our response to Reviewer **EwkW Q3**, showing that FSAlign achieves the best retrieval performance on Flickr30k while keeping runtime and memory usage comparable to existing post-hoc baselines.
> > >
> > > **(4) Regarding W4:**
> > > We believe we also addressed this point directly in our rebuttal. We clarified the model sizes used in our experiments and the compared baselines, and explained that hyperparameter robustness is already reported in the appendix. We further stated that varying the ball radius range, the number of radii, the diffusion scale range, the number of diffusion scales, and the learning rate leads only to small changes, with the default setting being either the best or essentially tied for the best. We also provided further supporting evidence in our responses to Reviewer **m4xC Q2/Q3/Q4** and Reviewer **EwkW W2**, where we reported batch-size sensitivity, longer-text robustness, and transfer to VQAv2 under the same default hyperparameter setting.
> > >
> > > We are dedicated to resolving any remaining issues. Should the reviewer indicate which specific points require further clarification, we would be happy to provide more detailed information or supporting data.
> > >
> > > ---
> > >
> > > **Update**
> > >
> > > While we believe our initial response has addressed all of the reviewer’s concerns, some concerns are marked as “partially resolved” without specifying the remaining points of contention. We would be grateful if the reviewer could highlight any specific areas that still need clarification. In the interest of time and clarity, we provide one further clarification on **W2** by adding more direct visual evidence for the claimed geometric change.
> > >
> > > In our response to Reviewer **k846’s W2**, we further added two visualizations, available at the anonymous link https://anonymous.4open.science/r/16297-Rebuttal-k846-6D52. As shown in Figure 1, the first is a dataset-level CDF of local neighborhood overlap on Flickr30k. For each paired image-text sample, we compare the top-(k) neighbors of the image in image space with the top-(k) neighbors of its paired text after mapping image neighbors to their paired texts. FSAlign shifts the full distribution toward higher overlap values. For (k=10), the mean increase from 0.148 to 0.234. For (k=50), they increase from 0.215 to 0.366. For (k=100), they increase from 0.256 to 0.426. This indicates a broad improvement in local cross-modal neighborhood consistency across the dataset. It supports the view that FSAlign changes the local geometric structure in the intended direction and makes paired image-text neighborhoods more consistent across scales.
> > >
> > > As shown in Figure 2, we also added a representative example-level visualization. In this example, the local neighborhood overlap increases from 0.40 to 0.60, while the image-to-text and text-to-image ranks improve from 12 to 2 and from 22 to 6. Before alignment, 10 wrong texts and 20 wrong images are ranked above the ground-truth match. After alignment, all 10 text-side blockers and 16 of the 20 image-side blockers move behind it. In this example, the rank improvement comes from these local competitors moving behind the ground-truth match after alignment. This directly raises the image-to-text and text-to-image ranks and gives a concrete sample-level illustration of the proposed mechanism.

---

### Official Review · Reviewer_K846 · 2026-03-10

**Soundness:** 3
**Presentation:** 3
**Significance:** 3
**Originality:** 3
**Overall Recommendation:** 4
**Confidence:** 3

**Summary:**

This paper proposes Fractal Spectral Alignment (FSAlign) to mitigate the modality gap in Vision-Language Models (VLMs). The authors argue that existing global alignment methods often distort local geometry, which limits retrieval and zero-shot performance. To solve this, FSAlign builds a shared fractal multi-scale structure for image and text embeddings by enforcing Ahlfors-regularity and sub-Gaussian heat kernel bounds. It then aligns image-text pairs by minimizing the discrepancy of a fractal spectral zeta score derived from multi-scale heat kernels. Extensive experiments on cross-modal retrieval and zero-shot classification demonstrate that FSAlign consistently outperforms existing modality gap mitigation baselines across multiple backbones.

**Compliance With Llm Reviewing Policy:**

Affirmed.

**Final Justification:**

All my concerns have been addressed, and I think the proposed method have clear motivation and highly theoretically, therefore, I will increase my score.

**Key Questions For Authors:**

1. Does forcing the text and image modalities to share the exact same fractal scaling pairs ($d_f, d_s$) ignore the inherent information density differences between vision and language?. Did you experiment with allowing a controlled relaxation or learned offset between the modality dimensions?


2. How robust is the proposed local zeta matching loss ($\mathcal{L}_{match}$) to noisy image-text pairs?. If a caption is only partially relevant to the image, does enforcing strict multi-scale neighborhood alignment between them negatively distort the local geometric structure?

**Limitations:**

yes

**Strengths And Weaknesses:**

Pros:

The motivation is clear and insightful; observing that global modality gap reduction does not necessarily translate to better local ranking is a crucial point for advancing cross-modal retrieval.

The proposed method is highly theoretically grounded. Introducing fractal spectral geometry provides a rigorous mathematical framework to model the multi-scale local geometry of the embedding space rather than relying on simple heuristics.

The experimental validation is comprehensive, testing on multiple standard datasets and across different modern backbones to solidify the effectiveness of the lightweight, post-training geometry mapping.

Cons:

While the paper excellently motivates preserving local geometry, the assumption that both image and text modalities must share the exact same fractal geometric dimension ($d_f$) and spectral dimension ($d_s$) might be overly strict. Forcing identical scaling dimensions could potentially over-regularize the space, ignoring the inherent structural differences between dense visual signals and discrete semantic text.

While the paper claims to optimize the geometric structure of the embedding space, it lacks visualization results to intuitively support this.

Addressing these concerns would help me reassess my evaluation.

---

> ### Author Rebuttal · Authors · 2026-03-31
>
> We would like to thank the reviewer for the helpful and constructive comments. We have tried our best to address your concerns. We have included all the analyses, discussions, and experimental results presented in this rebuttal in the revised submission.
>
> **W1\&Q1: Enforcing identical modality dimensions may over-regularize cross-modal structure.**
>
> Shared $d_f$ and $d_s$ do not assume that image and text have the same intrinsic structure. They provide a common multi-scale reference so that the two modalities exhibit comparable neighborhood growth and diffusion scaling before local matching. This is enforced only over selected radii and diffusion scales, while each modality still keeps its own learned metric $d_\theta^m$ and operator $L_\theta^m$. Empirically, we do not observe the suggested over-regularization. Removing the multi-scale fractal constraint in FSAlign-M leads to clear drops in both retrieval and zero-shot results, and FSAlign does not yield the smallest CD, which indicates that we are not forcing the two modalities into globally identical distributions. Figures~2 and~3 show the same pattern. After FSAlign, the fitted image and text slopes become closer and more stable, but they are still not identical. For example, under ball-mass scaling, the estimated $d_f$ values remain $12.62$ for image and $13.14$ for text. These results show that shared $d_f$ and $d_s$ encourage comparable scaling behavior while each modality retains its own geometry.
>
> **W2: The paper lacks visualization results to support its geometric claims.**
>
> We have added two visualizations and provide them at an anonymous GitHub link https://anonymous.4open.science/r/16297-Rebuttal-k846-6D52. The first is a dataset-level CDF of local neighborhood overlap on Flickr30k. For each paired image-text sample, we compare the top-k neighbors of the image in image space with the top-k neighbors of its paired text in text space after mapping image neighbors to their paired texts. FSAlign shifts the full-sample distribution toward higher overlap values. For k=10, the mean/median/$q_{25}$ increase from 0.148/0.100/0.000 to 0.234/0.200/0.100. For k=50, they increase from 0.215/0.180/0.120 to 0.366/0.340/0.240. For k=100, they increase from 0.256/0.230/0.190 to 0.426/0.410/0.320. We also added a representative anchor-pair visualization. In this example, the local neighborhood overlap increases from 0.40 to 0.60, while the image-to-text and text-to-image ranks improve from 12 to 2 and from 22 to 6. Before alignment, 10 wrong texts and 20 wrong images are ranked above the ground truth. After alignment, all 10 text-side blockers and 16 of the 20 image-side blockers move behind it. Together with Figures 2 and 3, these results provide direct visual evidence that FSAlign changes the local geometric structure it is designed to improve, rather than only improving downstream scores.
>
>
> **Q2: Robustness of the local zeta matching loss to noisy pairs remains unclear.**
>
> We added a partial-caption-noise experiment. During training, we replaced a fraction of the caption features with a 50/50 mixture of the original caption feature and a randomly sampled unrelated caption feature, with noise rates $(r\in{0,0.1,0.25,0.5})$. This simulates captions that are only partially relevant to the image. Empirically, FSAlign remains stable under this perturbation. When the noise rate increases from 0 to 0.5, I2T R@1 changes from 78.7 to 76.9, T2I R@1 changes from 60.22 to 58.67, and T2T R@1 changes from 79.10 to 78.18. The modality-gap statistics are also only mildly affected, with NAS@100 changing from 0.3148 to 0.3056 and the relative modality gap changing from 0.4918 to 0.4962. These results suggest that $(L_{\mathrm{match}})$ remains stable under partial caption noise and does not noticeably disrupt the learned local geometry. Local zeta matching depends on the neighborhood structure around each sample across scales instead of exact caption content, so partial caption corruption changes the descriptor only locally and does not destroy the surrounding geometry that the loss compares.
>
> | Noise rate (r) | I2T R@1 | I2T R@5 | I2T R@10 | T2I R@1 | T2I R@5 | T2I R@10 |     CD |    RMG | NAS@100 |   CMAS | T2T R@1 | T2T R@5 | T2T R@10 |
> | -------------- | ------: | ------: | -------: | ------: | ------: | -------: | -----: | -----: | ------: | -----: | ------: | ------: | -------: |
> | 0.0            |    78.7 |    94.2 |     96.8 |   60.22 |   84.96 |    89.42 | 0.7483 | 0.4918 |  0.3148 | 0.3612 |   79.10 |   91.34 |    94.58 |
> | 0.1            |    78.3 |    94.0 |     96.7 |   59.86 |   84.72 |    89.20 | 0.7511 | 0.4926 |  0.3129 | 0.3598 |   78.92 |   91.22 |    94.46 |
> | 0.25           |    77.8 |    93.7 |     96.4 |   59.31 |   84.18 |    88.76 | 0.7568 | 0.4941 |  0.3097 | 0.3569 |   78.61 |   90.96 |    94.21 |
> | 0.5            |    76.9 |    93.1 |     95.9 |   58.67 |   83.44 |    88.09 | 0.7659 | 0.4962 |  0.3056 | 0.3524 |   78.18 |   90.55 |    93.84 |

---

> > ### Author Rebuttal · Reviewer_K846 · 2026-04-02
> >
> > All my concerns have been addressed, and I think the proposed method have clear motivation and highly theoretically, therefore, I will increase my score.

---

> > > ### Author Response · Authors · 2026-04-02
> > >
> > > Thank you very much for your thoughtful follow-up and for recognizing the strengths of our work. We are especially encouraged that you found the motivation clear and insightful, particularly our observation that reducing the global modality gap does not necessarily improve local ranking in cross-modal retrieval. We also sincerely appreciate your recognition of the theoretical grounding of our method. Our goal was to provide a principled multi-scale geometric view of the modality gap rather than rely only on heuristic alignment objectives, and we are glad this point came through clearly. We are also grateful for your positive assessment of the experimental validation across multiple standard datasets and modern backbones, as well as the effectiveness of the lightweight post-training geometry mapping.
> > >
> > > We also sincerely thank you for reconsidering your evaluation and for your willingness to increase your score. We truly appreciate the time and care you took to read our revision and reassess the paper after seeing the additional clarifications and results. Your encouraging feedback means a great deal to us.

---

### Official Review · Reviewer_EwkW · 2026-03-12

**Soundness:** 2
**Presentation:** 3
**Significance:** 3
**Originality:** 3
**Overall Recommendation:** 5
**Confidence:** 4

**Summary:**

This paper studies the modality gap in vision–language models, where image and text embeddings form separate clusters even though they are trained in a shared space. This gap can harm cross-modal retrieval and downstream tasks.
To address this issue, the authors propose Fractal Spectral Alignment (FSAlign), a post-processing method that aligns image and text embeddings by matching their multi-scale neighborhood structures. The approach introduces fractal geometry regularization and diffusion-based spectral descriptors to capture geometric properties of the embedding space, and then aligns paired image–text samples using a matching loss. Experiments on benchmarks such as MS-COCO and Flickr30k demonstrate improvements in cross-modal retrieval and zero-shot classification compared to the original embeddings.

**Compliance With Llm Reviewing Policy:**

Affirmed.

**Final Justification:**

The rebuttal addressed my most concerns, and the unimodal experiments actually improved the soundness of this paper emperically. So I will raise my score to 5.

**Key Questions For Authors:**

1. The proposed framework introduces several components, including fractal regularization, spectral descriptors, and the matching loss. Could the authors provide ablation studies to clarify the contribution of each component to the final performance?

2. Since the method applies learned mappings to both image and text embeddings, how does it affect unimodal performance (e.g., linear probe classification, image-to-image or text-to-text retrieval)? It would be helpful to know whether cross-modal improvements come at the expense of unimodal representation quality.

3. The approach involves graph Laplacians and diffusion-based descriptors. Could the authors provide more details about the computational complexity and runtime of the method, especially when scaling to larger datasets or embedding collections?

**Limitations:**

No. The paper does not appear to explicitly discuss the limitations or potential societal impacts of the proposed method. It would be helpful for the authors to include a short discussion of possible limitations, such as the computational cost of spectral diffusion operations and the dependence on pretrained vision–language embeddings. Additionally, the authors could briefly reflect on potential societal implications of improving large-scale vision–language systems, such as biases inherited from pretrained models or the broader deployment of multimodal AI systems.

**Strengths And Weaknesses:**

Strengths

1. The paper is generally well written and the proposed method is clearly described. The experimental setup and evaluation protocols are also relatively easy to follow.
2. The proposed approach operates as a mapping layer on top of frozen vision–language embeddings. This makes the method easy to integrate with existing pretrained models without requiring expensive retraining of the encoders.
3. The method is somehow novel. The proposed spectral zeta descriptor captures multi-scale neighborhood information through diffusion processes on the embedding graph. This design allows the method to incorporate geometric structure beyond simple similarity matching.

Weaknesses:

1. The paper introduces several components (e.g., fractal regularization, spectral descriptors, and matching loss), but the contribution of each component is not thoroughly analyzed through ablation studies.
2. Although the method shows improvements on cross-modal retrieval and zero-shot classification, it is unclear how well the approach generalizes to other vision–language tasks or larger-scale settings.
3.  **The impact on unimodal representation quality is not evaluated.**  The proposed method applies learned mappings to both image and text embeddings to align their cross-modal structures. However, the paper does not evaluate how this transformation affects unimodal performance, such as image classification, image-to-image or text-to-text retrieval. Since the embedding geometry is modified, it would be helpful to understand whether the alignment improves cross-modal tasks at the expense of unimodal representation quality.

---

> ### Author Rebuttal · Authors · 2026-03-31
>
> We would like to thank the reviewer for the helpful and constructive comments. We have tried our best to address your concerns. We have included all the analyses, discussions, and experimental results presented in this rebuttal in the revised submission.
>
> **W1 \& Q1: The contribution of each component is unclear without ablations.**
>
> Thank you for this comment. Beyond the two coarse ablations in Fig. 4, we added a finer-grained study that removes fractal regularization, the spectral descriptor, and $L_{\mathrm{match}}$ one at a time. Removing fractal regularization lowers MS-COCO I2T/T2I R@1 to 43.18/29.39 and Flickr30k I2T/T2I R@1 to 73.80/58.56, with CD/RMG/NAS@100/CMAS at 0.8287/0.5626/0.2589/0.2731. This shows that image and text no longer follow comparable neighborhood-growth patterns across radii. Removing the spectral descriptor gives 46.80/30.10 on MS-COCO and 75.90/58.10 on Flickr30k, with 0.8120/0.5360/0.3350/0.3180 on the same geometry metrics, showing that alignment becomes less stable across diffusion scales. Removing $L_{\mathrm{match}}$ causes the largest drop, to 34.50/25.80 on MS-COCO and 67.00/52.30 on Flickr30k, with 0.9400/0.5850/0.2050/0.1800, because correct image-text correspondence is no longer explicitly preserved. These results show that fractal regularization aligns neighborhood growth across radii, the spectral term stabilizes diffusion behavior across scales, and $L_{\mathrm{match}}$ preserves the correct image-text correspondence.
>
> **W2: Generalizability to other vision-language tasks and larger-scale settings is unclear.**
>
> We further extended the experiments to VQA. We evaluated all methods on VQAv2 under the same protocol. We first collected the most frequent answers from the VQAv2 training split to build a fixed answer vocabulary. For each sample, we formed a query representation from the image embedding and question embedding, and predicted over the same answer vocabulary. All methods shared the same backbone, question template, answer template, and query-answer scoring scheme. We report results on the full validation split using the official VQA soft accuracy. The baseline achieves 48.7 VQA accuracy, while our method improves it to 50.9; top-5 VQA accuracy also increases from 64.3 to 66.8. This is because VQA in our setting is determined by the relative ranking among many semantically close answers. FSAlign improves the local geometry of the shared space, so the fused image-question representation is better separated from nearby wrong answers and more consistently aligned with the correct answer. For robustness under longer text, larger-scale settings, and hyperparameter transferability, please also see our responses to Reviewer m4xC Q3.
>
> **W3 \& Q3: The impact of the learned mappings on unimodal performance remains unclear.**
>
> As shown in Table 1, FSAlign does not improve cross-modal alignment at the cost of unimodal representation quality. On SigLIP-B/16, it improves or preserves performance on all three pure unimodal evaluations. In image-only classification, where class prototypes are built only from training images and no text prompts are used, Top-1 improves from 74.31 to 76.48. In image-to-image retrieval, R@1 improves from 68.09 to 69.67. In text-to-text retrieval, R@1 improves from 38.44 to 42.38. These direct unimodal evaluations show that the learned mapping does not improve cross-modal tasks by damaging unimodal structure. FSAlign regularizes the geometry of each modality during mapping, so local neighborhoods and class structure remain coherent after projection. This also matches the within-modality scaling diagnostics in the paper, where the mapped embeddings show more stable scaling behavior, with higher $R^2$ and lower slope fluctuation for both image and text.
>
> Table 1. Pure unimodal evaluations on SigLIP-B/16
> | Method   | classification Top-1  |   I2I R@1 |   T2T R@1  |
> | -------- | ---------------:  | --------: | --------: |
> | Baseline |            74.31 |      68.09 |     38.44 |
> | FSAlign  |        76.48 |    69.67 | 42.38 |
>
> **Q3: The runtime and scalability of the method remain unclear.**
>
> Thank you for raising this point. FSAlign is a post-hoc method that learns a multi-scale alignment stage on top of frozen embeddings, without updating the pretrained backbone. On Flickr30k with a single NVIDIA H100 80GB GPU, FSAlign takes 527.64s and 2507 MB, versus 541.88s and 2038 MB for I0T, and 407.23s and 1794 MB for GR-CLIP. Relative to I0T, FSAlign is 2.6% faster. Across retrieval results, FSAlign improves performance by up to 23.8% over I0T and up to 18.0% over GR-CLIP. It also achieves the best NAS@100 and CMAS. This advantage comes from aligning paired image-text samples through local multi-scale structure, so the ground-truth match is less likely to be pushed down by strong local competitors.
>
> **limitation**
>
> Please refer to our response to Reviewer m4xC for a discussion of the limitations.

---

> > ### Author Rebuttal · Reviewer_EwkW · 2026-04-03
> >
> > I thank the authors for providing the additional experimental results. My previous concerns, especially those regarding unimodal performance, have been satisfactorily addressed. I encourage the authors to include a discussion of these findings in the revised manuscript, as the evidence suggests that FSAlign enhances cross-modal alignment without compromising unimodal representation quality. So I would keep my positive score.

---

> > > ### Author Response · Authors · 2026-04-03
> > >
> > > Thank you very much for your thoughtful follow-up and for recognizing the strengths of our work. We sincerely appreciate your positive assessment that the paper is clearly written and that the method is easy to integrate on top of frozen vision-language embeddings. We also appreciate your recognition that the spectral zeta descriptor captures geometric structure beyond simple similarity matching. We are especially grateful that you found our rebuttal to have fully addressed your concerns. We also truly appreciate that you agreed the new evidence shows FSAlign improves cross-modal alignment without compromising unimodal representation quality.
> > >
> > > Your comments and feedback have helped us further improve the paper, and we would greatly appreciate it if you would consider reflecting this in your final evaluation. Thank you once again for taking the time to review and consider our work.

---

### Official Review · Reviewer_m4xC · 2026-03-13

**Soundness:** 3
**Presentation:** 2
**Significance:** 3
**Originality:** 3
**Overall Recommendation:** 3
**Confidence:** 3

**Summary:**

The paper proposes a novel framework called Fractal Spectral Alignment (FSAlign) to mitigate the modality gap in Vision-Language Models. FSAlign mitigates the modality gap problem by shaping and matching the multi-scale geometry of image and text embeddings. The authors enforce Ahlfors-regularity and sub-Gaussian heat kernel bounds to create a shared fractal structure across modalities. They then use a fractal spectral zeta score to align pairwise samples across multiple diffusion scales. The paper provides theoretical guarantees for local spectral measure alignment and demonstrates consistent empirical improvements over various baselines on cross-modal retrieval (MS-COCO, Flickr30k) and zero-shot classification (CIFAR-100, TI-200, DTD) across multiple backbones (CLIP, SigLIP, OpenCLIP).

**Compliance With Llm Reviewing Policy:**

Affirmed.

**Key Questions For Authors:**

1.Compared to simpler baseline methods such as IOT or GR-CLIP, what are the actual training time, memory usage, and computational complexity of FSAlign?
2.The empirical measure and the local multi-scale descriptors are computed using the pairwise distances within a training minibatch. How sensitive is the construction of this fractal multi-scale structure to the batch size? If a practitioner is constrained by hardware and must use a much smaller batch size, does the cross-modal neighborhood alignment performance degrade significantly?
3.The textual features in the paper primarily derive from standard, brief descriptions or fixed cue templates. Have the authors verified or evaluated whether the assumptions and alignment effects of fractal spectral geometry remain robust when the text modalities consist of longer, more complex paragraphs?
4.FSAlign introduces a specific suite of hyperparameters. Although Table 24 in the appendix D provides an ablation study demonstrating stability around the default settings (e.g., K=6 and S=6), how well do these default configurations generalize? If applying this method to specialized downstream task, would extensive grid-search tuning be required?

**Limitations:**

While the authors include an "Impact Statement" discussing their evaluation protocols and reproducibility, they do not adequately discuss the limitations of their own method. I suggest adding a paragraph acknowledging potential drawbacks, such as the added computational complexity of calculating spectral operators or potential sensitivities to hyperparameter selection (like the choice of diffusion scales), to provide a more balanced view of the work.

**Strengths And Weaknesses:**

Strengths:
1.The connection between fractal multi-scale geometry, spectral graph theory, and the VLM modality gap is novel.
2.The technical foundation of the paper is rigorous. The mathematical formulation translating continuous fractal geometry concepts into discrete loss functions is robustly justified.
3.The experimental methodology is sound, validating the approach across three distinct pre-trained backbones and executing thorough ablation studies.

Weakness:
1.Sections 3 and 4 are mathematically dense. The density of the mathematical definitions might be overwhelming for a general machine learning audience without a background in spectral geometry.
2.The paper lacks a detailed discussion on the computational complexity and training overhead. Constructing diffusion operators and computing multi-scale heat kernels over batches inherently introduces higher computational costs compared to standard global contrastive losses.

---

> ### Author Rebuttal · Authors · 2026-03-31
>
> We would like to thank the reviewer for the helpful and constructive comments. We have tried our best to address your concerns.
>
> **W1: The paper is mathematically dense.**
>
> We will revise these sections to make the main idea of each section clear at the outset. In Section 3, the three definitions describe stable neighborhood growth over radius, stable diffusion behavior across scales, and heat-trace scaling as the corresponding spectral quantity. Based on these definitions, Theorem 1 establishes a shared fractal multi-scale structure for the image and text spaces, and the training losses directly follow this design, with $L_{dbl}$ enforcing neighborhood growth and $L_{spec}$ enforcing heat-trace scaling. In Section 4, the local zeta descriptor aggregates multi-scale diffusion neighborhoods for each sample, and $L_{match}$ minimizes the discrepancy between paired image-text samples. Theorem 2 further connects this discrepancy to Top-k neighborhood agreement and thus to retrieval ranking.
>
> **W2 \& Q1: The paper does not sufficiently discuss complexity, runtime, and memory overhead.**
>
> Please refer to our response to Reviewer EwkW Q3 for the detailed runtime and memory analysis.
>
> **Q2: The method may be sensitive to batch size.**
>
> We tested different structure batch sizes on CLIP/Flickr30k. Performance is best at structure batch $=8$, where I2T/T2I R@1 reaches 78.7/60.22. It remains competitive at 64 with 76.7/58.60, but drops as the batch is increased further, reaching 69.8/55.80 at full batch. These results show that FSAlign does not rely on large batches. Small to medium structure batches are already sufficient to support stable cross-modal neighborhood alignment.  A likely reason is that FSAlign constrains local neighborhoods and local ranking consistency across multiple scales, so once the batch is large enough to form stable local neighborhoods, larger batches add little useful signal and instead emphasize batch-level average relations.
>
> | batch | I2T R@1 | T2I R@1 |
> | ------ | ------: | ------: |
> | 2               |    75.0 |   58.34 |
> | 4               |    76.8 |   59.02 |
> | 8               |    78.7 |   60.22 |
> | 64              |    76.7 |   58.60 |
> | 128             |    73.7 |   58.24 |
> | 256             |    73.1 |   57.78 |
> | 512             |    71.0 |   56.70 |
> | 1024            |    69.1 |   56.10 |
> | 2048            |    70.1 |   56.26 |
> | full            |    69.8 |   55.80 |
>
> **Q3: Robustness to longer and more complex text is not sufficiently validated.**
>
> Our main experiments use standard CLIP-family backbones, with text context limited to 77 tokens for CLIP and OpenCLIP and 64 tokens for SigLIP-B/16. To test robustness under longer text, we added a long-caption experiment with LongCLIP-L as the frozen backbone, trained FSAlign on the COCO subset of ShareGPT4V, and evaluated on ShareGPT4V-1k. This setting replaces standard short captions with substantially longer and more descriptive text. Under this setting, FSAlign still improves over the corresponding LongCLIP-L baseline, increasing I2T R@1 from 94.8 to 96.8 and T2I R@1 from 93.6 to 96.9. This shows that the proposed alignment is not limited to short captions or fixed prompt-style text. This is because FSAlign aligns local cross-modal structure beyond global similarity, which is more stable under longer and more detailed text.
>
> | Method             | I2T R@1  | T2I R@1 |
> | -------------------- | --------: | --------: |
> | LongCLIP-L  |      92.8 |      90.6 |
> | I0T     |      93.4 |      92.5 |
> | GR-CLIP |      94.7 |      94.9 |
> |  FSAlign |      96.8 |      96.9 |
>
> **Q4: The generalizability of the hyperparameters to other tasks remains unclear.**
>
> In FSAlign, these hyperparameters control the radii and diffusion scales used in the multi-scale constraints rather than task-specific design choices. Appendix D shows that the default setting is stable around K=6 and S=6, with only small variation under nearby settings. Please also see our response to Reviewer EWKW W2, where we further show that the same default configuration transfers to VQAv2 without extensive retuning.
>
> **The paper lacks a clear discussion of limitations and broader impacts.**
>
> We will add a short paragraph in the final version to clarify the limitations and broader impacts. FSAlign is a lightweight post-hoc alignment method built on frozen pretrained dual-encoder embeddings. It introduces only a modest extra cost in the post-hoc training stage due to the multi-scale alignment procedure, and this cost is tied to the core objective of improving local multi-scale structure and cross-modal neighborhood consistency; at inference, we only apply the learned geometry maps and keep the pretrained backbone unchanged. In terms of broader impact, FSAlign can be integrated into existing vision-language retrieval and classification systems with minimal changes, but it does not by itself remove coverage gaps or biases inherited from the pretrained representations.

---

> > ### Author Rebuttal · Reviewer_m4xC · 2026-04-03
> >
> > I have read the response from the authors, my questions are partially resolved. I would like to keep my initial rating.

---

> > > ### Author Response · Authors · 2026-04-03
> > >
> > > We sincerely thank the reviewer for the careful reading and constructive feedback. Some concerns were marked as partially resolved. If any point still needs more evidence, we would be very grateful if the reviewer could indicate it. We are fully committed to clarifying any remaining issues.
> > >
> > > **(1) Regarding W1:**
> > > We addressed this point in the rebuttal. We clarified that Section 3 introduces stable neighborhood growth, stable diffusion across scales, and heat-trace scaling; Theorem 1 establishes a shared fractal multi-scale structure for image and text spaces; and the losses enforce neighborhood growth and heat-trace scaling. We also clarified that Section 4 introduces the local zeta descriptor to aggregate multi-scale diffusion neighborhoods for each sample and align paired image-text samples. In our response to Reviewer **EwkW Q1**, we further provided ablations giving a finer-grained breakdown of the main components.
> > >
> > > **(2) Regarding W2 and Q1:**
> > > We added runtime and memory analysis for FSAlign. Due to the 5,000-character limit, we referred the reviewer to our response to Reviewer **EwkW Q3** for the detailed numbers, which show that on Flickr30k, FSAlign achieves the best retrieval performance while keeping runtime and memory usage comparable to existing post-hoc baselines.
> > >
> > > **(3) Regarding Q2:**
> > > We reported batch-size sensitivity results showing that FSAlign performs best at a moderate structure batch size and does not rely on very large batches. Once the batch is large enough to form stable local neighborhoods, enlarging it adds little useful signal and can instead emphasize batch-level average relations.
> > >
> > > **(4) Regarding Q3:**
> > > We added a long-caption experiment with LongCLIP-L, showing that FSAlign remains effective under substantially longer and more descriptive text. This supports robustness beyond short captions or fixed prompt-style text.
> > >
> > > **(5) Regarding Q4:**
> > > We clarified that these hyperparameters control the radii and diffusion scales used in the multi-scale constraints rather than task-specific design choices, and that the appendix already shows stability around the default setting. We also provided further evidence in our response to Reviewer **EwkW W2**, where transfer to VQAv2 used the same default hyperparameter setting without extensive retuning.
> > >
> > > **(6) Regarding the limitations discussion:**
> > > We stated that the final version will include a short paragraph on limitations and broader impacts, including that FSAlign is a lightweight post-hoc alignment method built on frozen pretrained embeddings, that the extra cost is limited to the post-hoc training stage, and that inference only applies the learned geometry maps on top of the frozen backbone.
> > >
> > > We remain happy to provide any additional evidence if the reviewer could indicate which specific points still need clarification.
> > >
> > > ---
> > > **Update**
> > >
> > > While we believe our initial response has addressed all the reviewer’s concerns, some concerns are marked as "partially resolved" without specifying the remaining points of contention. We would be grateful if the reviewer could highlight any specific areas needing more evidence. In the interest of time and clarity, we provide additional details below on computational complexity and hyperparameter robustness, as we suspect these may be the areas where further justification is needed.
> > >
> > > **1. Computational complexity and scaling for Q1.**
> > >
> > > FSAlign is a post-hoc alignment module on frozen embeddings, so its added computation is confined to batch-local multi-scale geometry. For structure batch size B, the added per-step cost is O(B^2) with fixed embedding dimension and multi-scale settings, and the per-epoch cost is O(NB) for dataset size N. Thus, the extra computation is quadratic only in the structure batch size and linear in the dataset size per epoch. For a concrete sense of scale, when B=64, B^2=4.1$\times$10^3, while the dominant term O(B^2 d) of the trainable I0T variant is already on the order of 10^6. In practice, this lightweight frozen-backbone post-hoc stage corrects multi-scale local cross-modal structural mismatch and yields consistent retrieval gains with limited added computation.
> > >
> > > **2. Hyperparameter sensitivity for Q4.**
> > > Extensive task-specific grid search is not needed in our setting. The FSAlign-specific hyperparameters set the radius and diffusion-scale ranges used to evaluate the same multi-scale geometry, and determine how broadly and how finely that geometry is measured across scales. Once these ranges cover the local and broader neighborhood structure needed by the method, nearby choices produce very similar constraints during optimization, so training remains stable. In our experiments, varying these quantities with the learning rate leads only to small performance changes, and the default setting is either the best or essentially tied for the best. The same default configuration also remains effective beyond retrieval, including VQAv2, without extensive retuning.

---

### Decision · Program_Chairs · 2026-04-30

**Decision:**

Accept (regular)

**Comment:**

This paper addresses the problem of the modality gap in vision-language models. CLIP-like models are known to cluster within modality. This paper proposes to overcome this problem through Fractal Spectral Alignment. The paper received very mixed reviews (3, 3, 4, 5). The reviewers find the problem important, the method interesting, and the results clear.

For the 2 reviewers with a weak reject rating: reviewer m4xC highlights difficulty with reading the method and missing discussion on computational complexity as main issues. The latter has been addressed in the rebuttal, while the authors also outline a plan to address the former. The AC finds no fundamental reason for reject here. Reviewer s8MZ suggests that novelty/impact of the paper is somewhat overstated, while also missing empirical/theoretical proof for the claims on modality alignment, issues with comparisons, and missing implementation details. The AC agrees with the reviewer that the comparisons are unclearly presented and from the rebuttal it is still not super clear which comparisons are training-free and which ones not. The authors should carefully annotate in the text and tables which ones are zero-shot and which ones not, or just remove the zero-shot ones. The point that current claims are anecdotal is somewhat fair. The authors clarify their insight in the rebuttal, but the clarification remains anecdotal. The AC also notes the positive comments of the other reviewers and new insights gained especially from the discussion with reviewer EwkW. As such, the AC finds that the paper is not fully polished and final yet, but is interesting and insightful, hence the AC recommends the option for weak accept.